# CIRCUIT HMM: A DETERMINISTIC HIDDEN MARKOV MODEL FOR AUTOMATED SEQUENTIAL CIRCUIT DESIGN

## ABSTRACT

Designing logic circuits requires significant time for manual programming, which hinders the rapid iteration of product development. To alleviate this extensive manual effort, researchers have investigated machine learning methods for automatically programming the hardware description language (HDL) code, and have achieved success in designing combinational circuits. However, due to the complexity of internal state transitions, the design accuracy for sequential circuits remains insufficient for practical applications, whereas the design accuracy of combinational circuits can achieve 99.99999999999%.

This paper proposes a novel machine learning model, Circuit HMM, a deterministic Hidden Markov Model (HMM), for accurately designing sequential circuits. Our key insight is that the input-output relationship of the sequential circuits can be formalized as a Markov Process, significantly reducing the design space. With this computationally efficient model, we prove that the design accuracy of the sequential circuit converges to 100% with linear complexity. Circuit HMM (1) first learns the hidden states by constructing an effective finite state machine (FSM) by heuristic **state mining**, which ensures the error rate caused by the inaccurate states converges to zero; (2) accurately transforms the sequential circuit design problem into a series of combinational circuit design problems by efficient **state encoding**; and (3) then learns the combinational circuit implementation from the input-output relations with a state-of-the-art logic regression tool, i.e. the BSD Learner, which ensures the combinational error rate converges to zero. Experimental results demonstrate that the proposed method can accurately design real-world circuit modules comprising up to 5,000 logic gates, significantly outperforming the state-of-the-art. In 41 out of 43 cases, the design accuracy converges to 100% within 5 minutes.

## 1 INTRODUCTION

The massive human design effort involved in designing logic circuits Bentley (2005) hinders rapid product iteration in microchips, necessitating automated methods to program Hardware Description Languages (HDL) code without human intervention. Recent machine learning advancements, including Large Language Models (LLMs) Liu et al. (2024); Xiao et al. (2024); Vijayaraghavan et al. (2024), Graph Neural Networks (GNNs) Zhang et al. (2019), Reinforcement Learning (RL) Roy et al. (2021); Zuo et al. (2023), and Symbolic Machine Learning Models such as Binary Speculation Diagram Learner (BSD) Cheng et al. (2024a;b) are applied to design HDL codes automatically.

The logic circuits can be categorized into two types: the combinational circuits and the sequential circuits. Recent successes primarily focus on combinational circuits, but state-of-the-art methods struggle to accurately design sequential circuits of similar scale, as they cannot effectively model the vast number of possible transitions in the extensive sequential space. As shown in Figure 1 (a) and (b), the output-port signals (also known as Primary Outputs, *PO*) at a specific time step $t_0$ can be related to the input-port signals (also known as Primary Inputs, *PI*) in any time step from the very start to this moment. The long-term complex relations make state-of-the-art methods ineffective.

To alleviate the design complexity, we observe that the input-output relation of the sequential circuits can be formalized as a Markov Process, i.e., there exists a set of internal states capturing all the relative history information; the output and state transition of the circuit are only affected by

Figure 1: **The basic structure of a sequential circuit and its HMM-based modeling.** (a) A sequential circuit to be designed. (b) A traditional modeling of the sequential circuit, in which output depends on its present input signals and on the sequence of past inputs, i.e., the input history. (c) A Hidden Markov model of the sequential circuit, in which the output depends only on its present input and the present state. (d) A sequential circuit representation based on the Hidden Markov model.

the latest input and these internal states. This Markov Assumption for the sequential circuits significantly simplifies the design process within the vast sequential space. Specifically, the HMM model unfolds the sequential circuits into two combinational components: a Transition Function to control the transition of the internal states and an Output Function to control the sequential circuit outputs. As shown in Figure 1(c), the design space of the sequential circuit is significantly reduced(see Appendix A.3.1), as it is only related to the latest time step with both the Transition Function (in green, representing the internal states functionality) and the Output Function (in blue, representing the output port functionality). If both combinational components can be automatically designed with machine learning methods, they can combine to build the corresponding sequential circuit straightforwardly, as shown in Figure 1(d).

However, sequential circuit design is a significantly more challenging task than solving conventional Markov processes due to its strict accuracy requirements. It cannot be solved by either the classical Hidden Markov Models or the state-of-the-art machine learning methods, including LLMs Liu et al. (2024); Xiao et al. (2024); Vijayaraghavan et al. (2024), NNs Zhang et al. (2019), Reinforcement Learning Roy et al. (2021); Zuo et al. (2023), and Symbolic Models Cheng et al. (2024a;b). The reason is that the input-output relation of the sequential circuit is a deterministic type of Markov Process, where, given a specific input sequence after resetting the circuit, the output sequence remains unchanged. This feature imposes a much stricter accuracy requirement, where not only do we need to learn the probability distribution of the process, but also to determine the precise Boolean function that relates inputs to outputs. As a result, classical Hidden Markov Models fail to design sequential circuits because their Transition Function and Output Function are represented in probabilistic matrices, which do not effectively model the deterministic system. State-of-the-art machine learning methods lack a concrete extraction of the internal hidden states, which do not effectively model the sequential behavior of the circuit. In this case, these methods cannot achieve the high design accuracy required, for example, $> 99.9999\%$, even on small-scale sequential circuits with only hundreds of logic gates.

We propose **Circuit HMM**, a deterministic Hidden Markov Model for accurate automated sequential circuit design. This model works in three steps: it (1) first learns the hidden states by constructing an effective finite state machine (FSM) by heuristic **state mining**, which ensures the error rate caused by the inaccurate states converges to zero; (2) accurately transforms the sequential circuit design problem into a series of combinational circuit design problems by efficient **state encoding**; and (3) then learns the combinational circuit implementation from the input-output relations with a state-of-the-art logic regression tool, i.e. the BSD Learner, which ensures the combinational error rate converges to zero. In the first step, the hidden states are learned by heuristic state mining, which includes both state exploration and state validation. State exploration utilizes Breadth-First Search (BFS) to explore the potential internal states, and state validation employs a Monte Carlo-based state validation method to determine whether the newly explored state is an undiscovered one. With these two steps, we prove that as more internal states are discovered from the circuit, the error rate caused by inaccurate internal states reduces monotonically and converges to zero. In the second step, after all the necessary states are determined, state encoding accurately separates the sequential circuit design problem into several combinational circuit design problems. It utilizes a simulated annealing algorithm to compress the similar combinational logic of each state, thereby eliminating the need to learn these combinational logic parts multiple times. In the third step, all the combinational circuits are learned by a state-of-the-art logic regression tool, i.e., the BSD Learner, which is proven to

achieve an arbitrarily small error bound with sufficient input-output examples Cheng et al. (2024a). When all the combinational circuits are accurately learned, the sequential circuit is determined, and its HDL code can be generated automatically.

The proposed method is evaluated on the widely used *VerilogEval v2* benchmark Liu et al. (2023); Pinckney et al. (2024) and circuit modules from the commercial DesignWare IP dataset by Synopsys Synopsys (2025). Experimental results demonstrate that the proposed method can accurately design real-world circuit modules comprising up to 5,000 logic gates, significantly outperforming the state-of-the-art. In 41 out of 43 cases, the design accuracy converges to 100% within 5 minutes.

In this paper, we make the following contributions:

- In automated sequential circuit design, we observe that formulating the input-output relations of the sequential circuit as a Markov Process significantly reduces the design space.

- We propose **Circuit HMM**, a deterministic HMM for accurate automated sequential circuit design. With this computationally efficient model, we prove that the design accuracy of the sequential circuit converges to 100% with linear complexity.

- Experimental results demonstrate that the proposed method can accurately design real-world circuit modules comprising up to 5,000 logic gates, significantly outperforming the state-of-the-art. In 41/43 cases, the design accuracy converges to 100% within 5 minutes.

## 2 PRELIMINARY

In this section, we present prior knowledge relevant to our problem, i.e., automated sequential circuit design. First, we introduce the problem definition, along with its historical and recent developments. Second, we introduce its machine learning background and our related works in this area.

### 2.1 AUTOMATED CIRCUIT DESIGN

Designing logic circuits automatically began to gain the attention of the academic community in the very early days of computer science and artificial intelligence. Alonzo Church proposed the "Church's synthesis problem" in the 1950s Church (1957) to design sequential circuits from input-output examples. We use the exact definition of automated circuit design as in previous papers Cheng et al. (2024a): Automated logic design is to generate the circuit logic with only the input-output examples of a specific sequential circuit, such that the design accuracy converges to 100%.

**Definition 1** (Automated Sequential Circuit Design). *There is a sequential logic oracle $\phi$ : $\{0,1\}^n \mapsto \{0,1\}^m$, which can only be observed by the input-output examples. By actively sampling at most $N$ input-output probes $\{(\mathbf{x}_1, \phi(\mathbf{x}_1)), (\mathbf{x}_2, \phi(\mathbf{x}_2)), \ldots, (\mathbf{x}_N, \phi(\mathbf{x}_N))\}$ from the oracle, where $\mathbf{x}_i, \phi(\mathbf{x}_i)$ is the input/output sequence, generate a sequential circuit logic $\psi$ to simulate $\phi$, such that $\forall \mathbf{x} \in \{0,1\}^n$,*

$$P(\phi(\mathbf{x}) = \psi(\mathbf{x})) \geq 1 - \epsilon \ (\epsilon \to 0),$$

*where the sequential circuit $\psi$ is a valid output design.*

### 2.2 SEQUENCE MODELING

According to the definition above, automated sequential circuit design belongs to the classical sequence modeling problem in machine learning, while it requires much stricter accuracy requirements. Sequence modeling tasks involve capturing the dependencies and patterns within sequential data, such as hidden state transitions. One widely adopted framework is the Input-Output Hidden Markov Model (IOHMM) Bengio & Frasconi (1996), which extends traditional Hidden Markov Models (HMM) by incorporating input-output dependencies. This makes it suitable for sequence prediction tasks. It is a probabilistic graphical model designed for simple sequence tasks, which has discrete hidden states and models input-output relationships using conditional probabilities. LSTM Hochreiter & Schmidhuber (1997) replaced IOHMM's discrete states with continuous hidden states and used neural networks to model nonlinear dependencies. This made LSTM suitable for more complex sequence tasks and long-term dependencies. Transformer Waswani et al. (2017) introduced self-attention mechanisms to replace recurrence, enabling parallel computation and efficient modeling of global dependencies. However, the probabilistic nature of the above methods introduces uncertainty in the outputs, which is undesirable for hardware design tasks that require deterministic results and cannot achieve the strict accuracy requirement. See more details in Appendix A.11.

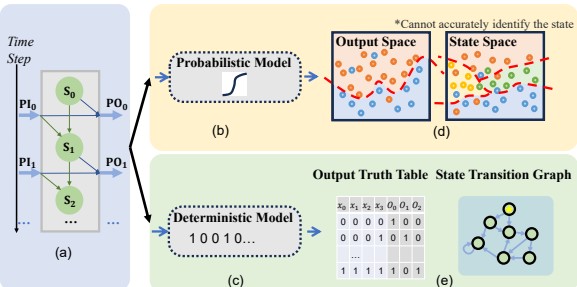

Figure 2: **Two Types of Hidden Markov Model (HMM) Solvers for Circuit Design.**

State-of-the-art deterministic sequence modeling methods are employed to learning the state structures of automata Steffen et al. (2012), but they cannot be used to design sequential circuits due to their exponential complexity. Approaches such as the L* algorithmAngluin (1987); Tappler et al. (2019) and the L# algorithmVaandrager et al. (2022) can capture the state transitions of the automata. However, the output function of these methods is the truth tables, whose representative complexity is exponential and impossible to implement on hardware. As a result, these methods can only be used in the state mining process but fail to design sequential circuits end-to-end.

## 3  HMM-BASED SEQUENTIAL CIRCUIT MODELING

In this section, we delve deeper into our key idea, i.e., modeling the automated sequential circuit design problem with its Markov property and solving it with our proposed deterministic HMM solver. First, we discuss the Markov property of the sequential circuits and how to model it with HMM models. Second, we argue that the current HMM solvers are all probabilistic, making it difficult to achieve a high accuracy, e.g. $> 99.99\%$. More specifically, we categorize the HMM solvers into two groups: probabilistic and deterministic, and provide an overview of the backgrounds. We further demonstrate that the deterministic ones are superior to the probabilistic ones.

**Markov Property of Sequential Circuit.** Sequential circuits contain hidden internal states, with $PO$ typically dependent on both external inputs and internal states (Figure 1(a)), posing challenges for automated design. First, in contrast to the one-to-one mapping of $PI$ to $PO$ in combinational circuits, the internal states of sequential circuits result in one-to-many mappings between $PI$ and $PO$, making conventional automated combinational circuits methods ineffective. Second, although $PO$ can be determined by the $PI$ if with the assumption of a fixed initial state, the circuit generation is complicated by significant complexity. In this case, the circuit either holds $O(T)$ $PI$ ports or $O(T)$ inference complexity, causing infeasible hardware cost and inference delays under vast sequence length $T$ (Figure 1(b)). Observing that the $PO$ depends only on the current internal state and $PI$, we model the sequential circuit as a Markov process to eliminate the need to maintain long input sequence information. Although the internal state is hidden, the Markov property allows us to maintain it implicitly or explicitly, decoupling the circuit generation into the transition function $LO = F_{trans}(PI, LI)$ and the output function $PO = F_{out}(PI, LI)$, where $LI$ (short for latch input) represents current state and $LO$ (short for latch output) are used to update the next-step state. This decoupling removes the time or area complexity caused by sequence length(Figure 1(c)). Further, by connecting the $LO$ of the transition function to the $LI$ ports in both functions through external latches, we construct the targeted sequential circuit(Figure 1(d)).

**Probabilistic and Deterministic HMM Solvers.** To address the state-agnostic problem of Markov sequential circuits, there are two types of solvers (Figure 2(b,c)): probabilistic and deterministic methods. Probabilistic methods implicitly maintain an uncountable number of internal states and determine output values based on the associated probabilities. LSTM is representative of this approach, maintaining and updating internal cell states with input at each step. However, probabilistic solvers struggle to meet the demands for high accuracy and low hardware cost. On the one hand, the typical coupling and inseparability of feature (state) and output spaces in probabilistic models (Figure 2(d)) can lead to cumulative errors, making it difficult to achieve high accuracy. On the other hand, these models require a significant parameter cost to learn even simple functions, resulting in substantial hardware redundancy. Compared to probabilistic solvers, deterministic solvers utilize explicit state representation and precise function generation, potentially avoiding accuracy errors and hardware redundancy. Since the states of real-world sequential circuits need to be fi-

nite for manufacture, we can regard the sequential circuit as FSM (Figure 2(e)). In this paper, we propose state mining and precise discrete state encoding methods to convert the sequential circuit generation problem into combinational circuit generation problems and integrate the state-of-the-art deterministic solver (i.e., the BSD Learner) to ensure accuracy.

## 4 METHODOLOGY

Circuit HMM utilizes the Markov assumption, allowing the output to rely solely on the internal state and the most recent input data as a combinational logic. In this way, for two specific internal states, if all the outputs are the same for every infinite input data trace, these states can be considered equivalent states. Furthermore, if there exists a state that is not equivalent to any of the internal states in the sequential circuit, the circuit design is not accurate. In this way, the accuracy of the sequential circuit design can be considered two-fold: the accuracy of the internal states, and the accuracy of the combinational logic after the internal states are determined.

Circuit HMM consists of three main stages as shown in Figure 3: **State Mining**, **State Encoding**, and **Circuit Generation**. (1) **State Mining** ensures the error rate caused by the inaccurate states converges to zero. We transform the state mining problem in sequential circuits into a BFS traversal problem on FSM state graphs (Figure 3(a)). Starting from the initial state, we iteratively explore reachable states and validate whether the explored state is necessary based on its $PO$ sequence comparison. (2) **State Encoding** accurately transforms the sequential circuit design problem into a series of combinational circuit design problems. We explicitly encode states for our combinational circuit generator (BSD-learner) and propose an SA-based state encoding optimization method to reduce the size of the generated circuits (Figure 3(b)). (3) **Circuit Generation** ensures the combinational error rate converges to zero. We generate combinational transition and output circuits based on explicit state encoding and connect the $LI$ and $LO$ using external latches to construct the final sequential circuit (Figure 3(c)). We provide a brief example of the state mining process in Appendix A.1.

### 4.1 STATE MINING

The goal of State Mining is to determine the necessary internal states for the sequential circuit from the input-output examples, thereby avoiding design errors caused by inaccurate internal states. We propose an iterative method to explore and check reachable states of the sequential circuit, thereby representing the sequential behavior of the target circuit with a finite set of necessary states, i.e., constructing an FSM representation. The state mining process is illustrated in Figure 3(a), and we provide a pseudocode in Appendix A.2. In every iteration, the method is a two-step process, comprising state exploration and state validation. In state exploration, it explores another reachable state from the current state set. In state validation, it determines whether the explored state is necessary and should be added to the current state set.

**State Exploration:** At the beginning of each mining iteration, we start from the set of currently known states and randomly generate a set of $PI$ to explore all potential reachable states. According to the state transition, we randomly sample the next state from the set of reachable states, which means that the state can be transitioned to from the current state with the current input. These randomly generated $PI$ sequences drive the circuit into potential new states, and the corresponding $PI$ sequences record the paths from the initial states to these potential states.

Thus, the state exploration process is transformed into a graph traversal problem based on Breadth-first Search (BFS). First, the reset state is pushed into a queue $Q(s)$ for known new states. Then, during each iteration, a known state is populated and is used to obtain the newly discovered potential states as described above. Each of those potential states is validated to determine whether it is a new state. Once successfully validated, they are pushed into $Q(s)$, and the corresponding $PI$ sequences are stored. In the next iteration, another known state is populated from $Q(s)$, and we randomly generate new $PI$ sequences starting from the state to explore all potential states at the next time step. This process continues until the entire state space is covered.

We prove (in Appendix A.3.2) that the expectation of the probability mass of undiscovered states satisfies:

**Lemma 1.** *For any finite state set $S^*$, the expectation of finding undiscovered states $p(s)$ after $N$ samples satisfies:*

$$\mathbb{E}[\sum_{s \notin S_N} p(s)] \leq \frac{1}{N}. \tag{1}$$

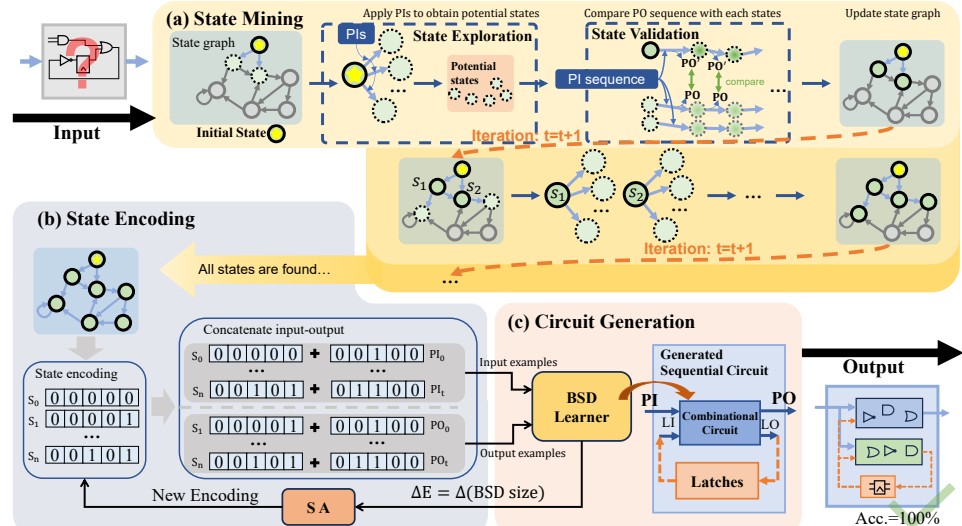

Figure 3: **Method Overview.** (a) *State mining:* At each iteration, explore potential states reachable from all known states in a single step. Validate each potential state to determine whether it is a new state. If a new state is found, update the state graph accordingly. (b) *State encoding:* Encode each state $(S^0, S^1, ..., S^n)$ to $LI$ and $LO$ in the state graph and concatenate $(PI, LI) / (PO, LO)$ as the inputs to the BSD Learner. Then, optimize the state encoding using the simulated annealing (SA) algorithm. (c) *Circuit generation:* Generate a combination circuit design, and add a latch feedback loop to convert the circuit into the HDL code of the sequential circuit.

where $S_N$ denotes the set of states that are discovered after $N$ samples.

This implies that the accuracy of the state mining process can converge to 100% with more examples.

**State Validation:** In this step, we determine whether each potential state is a new state by comparing its $PO$ behavior with known states. Since the state value is hidden, we compare several $PO$ sequences generated under an identical $PI$ sequence to determine whether a potential state is new.

The process is as follows: Firstly, starting from the potential state $S_p$ and each known state $S_k$, we apply the same $PI$ sequence and record the $PO$ generated at each time step as the $PO_p$ sequence and the $PO_k$ sequence. We repeat the above process for $M_{seq}$ times. Secondly, we compare each $PO_p$ sequence with all $PO_k$ sequences in the known state set. If every $PO_p$ sequence is identical to every $PO_k$ sequence of a known state, the potential state is considered equivalent to that known state. If $PO_p$ sequence is different from all $PO_k$ sequences, the potential state is regarded as a new state. It is added to the known state set, i.e., the specific $PI$ sequence leading to that state is saved. This iterative comparison process ensures that the known state set is updated dynamically and remains compact.

It is unrealistic to exhaust the infinite design space of the input data sequence. Therefore, we propose a Monte Carlo-based state validation method and prove (in Appendix A.4) that the expectation of the validation error converges to zero as the sequence length increases. In practice, we propose a Sequence Length Estimation Method in Appendix A.4, which demonstrates that for real-world circuit modules, the sequence length can be computed by an experimental function, varying from 100 to 5000 in our dataset.

**Theorem 1.** *Given two circuit internal states $S_i$ and $S_j$, there exists an integer T, such that for $\forall \epsilon \to 0$ and a random input sequence $PI_{seq}$ with length T, if the output sequences are validated, i.e., $PO_{seq}(S_i) = PO_{seq}(S_j)$. The expectation of the design error $E(S_i \not\equiv S_j) < \epsilon$.*

### 4.2 STATE ENCODING

In this stage, State Encoding accurately transforms the sequential circuit design problem into a series of combinational circuit design problems. We explicitly encode states for our combinational circuit generator (BSD-learner) and propose an SA-based state encoding optimization method to reduce the size of the generated circuits. The State Encoding process is divided into two main parts: 1) Assign

binary codes to the states based on the number of known states and their transitions, and derive the state transition relationships. This results in binary representations of the state transition function and the circuit functionality. 2) Use the BSD node count ($N_{BSD}$) as the optimization objective to refine the encoding, aiming to design a more compact circuit.

We use the minimum number of bits ($\lceil log_2 N \rceil$ bits) to represent all states ($N$ states) to design a compact circuit. We assign incremental binary codes to the states in the order they are added to the known states set. These binary codes are referred to as $LI$ (Latch Input), representing the current state of the circuit. Furthermore, $LO_t$ (Latch Output) is also regarded as $LI_{t+1}$ and can be determined by encoding matching with state validation. Once the state encoding and transition relationships are obtained, we concatenate the state encoding with the model's original inputs ($PI$) and outputs ($PO$). The BSD Learner then designs two circuits: Transition Circuit $LO = C_{trans}(PI, LI)$ and Output Circuit $PO = C_{out}(PI, LI)$, where $C_{trans}$ and $C_{out}$ are the circuit functions generated by the BSD Learner. Simultaneously, the BSD Learner evaluates the circuit size by counting the number of BSD nodes ($N_{BSD}$).

We optimize the state encoding using a Simulated Annealing (SA) algorithm, with $N_{BSD}$ as the objective function. The binary encoding of states impacts the gate counts of the generated circuits, presenting a large optimization opportunity. To address the large search space ($O(2^n!)$), we propose SA-based state search methods. At each iteration, a new solution is generated by randomly swapping the binary codes of two states. If the new solution results in a smaller $N_{BSD}$, it is accepted as the current solution; otherwise, it is accepted with a certain probability of encouraging exploration. Based on the temperature adjustment configurations, we propose two SA-based methods: (1) *Continuous SA*: It features a slower temperature decay rate, encouraging sustained convergence after initial exploration to yield better encoding quickly. (2) *Cyclic SA*: To avoid being trapped in local minima, it periodically raises the temperature to balance global exploration and optimization to find the global optimum over a long period. This is achieved by setting a larger decay rate and ensuring the $Temperature$ is reinitialized to a higher value once it drops below a predefined threshold.

## 4.3 CIRCUIT GENERATION

After State Mining and State Encoding, the sequential circuit design problem has been transformed into a series of combinational circuit design problems. In this stage, circuit generation consists of two main steps: 1) generating the combinational circuits using the state-of-the-art method, i.e., the BSD Learner, and 2) outputting the corresponding HDL code of the sequential circuit to be designed.

Once an optimized state encoding is obtained, we use the BSD Learner to solve the state transition function $LO = F_{trans}(PI, LI)$ and the functionality function $PO = F_{out}(PI, LI)$. To transform the combinational circuit into the output HDL code of a sequential circuit, it is essential to ensure that the output state $LO$ serves as the input state $LI$ in the next time step. This is achieved by introducing a latch loop between $LO$ and $LI$. As a fundamental storage element in sequential circuits, latches hold input information and provide it as output in the next time step. By adding a latch loop, we connect $LO$ to the input of the latch and the output of the latch to $LI$. This operation does not modify the internal structure of the existing combinational circuit. Instead, it ensures the correct sequential behavior by maintaining the state transition relationship across time steps.

Through this modification, the BSD-learner-generated combinational circuit is converted into a sequential circuit with inputs $PI$ and outputs $PO$. Theorem 2 provide a guarantee to ensure that with enough samples, the expectation of the error rate can be converged to 0. The proof of this theorem is in Appendix A.3.3.

**Theorem 2.** *For any $\epsilon > 0$, there exists a $N$ such that for all $N > \frac{1}{\epsilon}$, the expectation of the error rate $E(N)$ of our learned model $C$ satisfies:*

$$\mathbb{E}[E(N)] < \epsilon. \tag{2}$$

*where $N$ is the length of the state mining $PI$ sequence with reset.*

## 5 EXPERIMENT

**Datasets.** We evaluate all methods on the *VerilogEval v2* dataset Liu et al. (2023), a widely used benchmark in automatic circuit design, and the DesignWare IP library Synopsys (2025), a most commonly used circuit set in the front-end EDA flow covering categories such as control logic, arithmetic components, and registers.

Table 1: **Comparison of sequential circuit generation in terms of accuracy and circuit size.** Our method achieves the highest accuracy while maintaining a significantly small circuit size. Due to space limits, simple circuits with fewer than 5 states and 30 gates are not shown here.

| Circuits | # States | Acc.(%) | | | | Circuit Size(# gates) | | | |
|---|---|---|---|---|---|---|---|---|---|
| | | Prob | Deterministic | Prob+HMM | Ours | Prob | Deterministic | Prob+HMM | Ours |
| Prob041 | 256 | 32.89% | 49.65% | **100.00%** | **100.00%** | $3.12 \times 10^{12}$ | 8 | $1.17 \times 10^8$ | 48 |
| Prob046 | 256 | 29.65% | 50.09% | **100.00%** | **100.00%** | $3.12 \times 10^{12}$ | 8 | $1.17 \times 10^8$ | 48 |
| Prob047 | 256 | 32.09% | 50.00% | **100.00%** | **100.00%** | $3.12 \times 10^{12}$ | 8 | $1.17 \times 10^8$ | 48 |
| Prob067 | 10 | 27.56% | 62.13% | **100.00%** | **100.00%** | $3.11 \times 10^{12}$ | 1 | $1.04 \times 10^8$ | 60 |
| Prob085 | 16 | 34.27% | 56.12% | **100.00%** | **100.00%** | $3.11 \times 10^{12}$ | 9 | $1.12 \times 10^8$ | 78 |
| Prob096 | 5 | 99.88% | 1.88% | **100.00%** | **100.00%** | $3.11 \times 10^{12}$ | 1 | $1.03 \times 10^8$ | 33 |
| Prob121 | 5 | 56.72% | 49.32% | **100.00%** | **100.00%** | $3.11 \times 10^{12}$ | 1 | $1.03 \times 10^8$ | 36 |
| Prob133 | 8 | 83.54% | 87.31% | **100.00%** | **100.00%** | $3.11 \times 10^{12}$ | 2 | $1.05 \times 10^8$ | 54 |
| Prob136 | 6 | 73.29% | 74.27% | **100.00%** | **100.00%** | $3.11 \times 10^{12}$ | 1 | $1.03 \times 10^8$ | 33 |
| Prob137 | 12 | 96.44% | 95.84% | **100.00%** | **100.00%** | $3.11 \times 10^{12}$ | 1 | $1.03 \times 10^8$ | 57 |
| Prob138 | 6 | 59.85% | 74.99% | **100.00%** | **100.00%** | $3.11 \times 10^{12}$ | 1 | $1.03 \times 10^8$ | 39 |
| Prob140 | 10 | 97.16% | 99.48% | **100.00%** | **100.00%** | $3.11 \times 10^{12}$ | 1 | $1.04 \times 10^8$ | 81 |
| Prob146 | 2303 | 95.58% | 97.68% | 99.84% | **100.00%** | $3.13 \times 10^{12}$ | 1 | $1.05 \times 10^8$ | 2181 |
| Prob149 | 6 | 51.60% | 36.96% | **100.00%** | **100.00%** | $3.11 \times 10^{12}$ | 3 | $1.07 \times 10^8$ | 96 |
| Prob152 | 6 | 47.84% | 58.68% | **100.00%** | **100.00%** | $3.11 \times 10^{12}$ | 4 | $1.09 \times 10^8$ | 81 |
| Prob155 | 88 | 48.22% | 58.57% | 99.99% | **100.00%** | $3.11 \times 10^{12}$ | 4 | $1.09 \times 10^8$ | 357 |
| bictr_scnto | 256 | 51.33% | 55.59% | 99.62% | **100.00%** | $3.12 \times 10^{12}$ | 11 | $1.22 \times 10^8$ | 189 |
| lfsr_load | 256 | 28.78% | 49.78% | 50.62% | **100.00%** | $3.12 \times 10^{12}$ | 10 | $1.20 \times 10^8$ | 597 |
| lfsr_scnto | 256 | 24.64% | 55.92% | 94.47% | **100.00%** | $3.12 \times 10^{12}$ | 10 | $1.20 \times 10^8$ | 129 |
| lfsr_updn | 255 | 35.24% | 65.12% | 64.61% | **100.00%** | $3.11 \times 10^{12}$ | 2 | $1.07 \times 10^8$ | 879 |
| reg_s_pl | 256 | 79.88% | 83.18% | **100.00%** | **100.00%** | $3.11 \times 10^{12}$ | 10 | $1.20 \times 10^8$ | 96 |
| shftreg | 16 | 34.94% | 50.30% | **100.00%** | **100.00%** | $3.11 \times 10^{12}$ | 7 | $1.14 \times 10^8$ | 66 |
| updn_ctr | 256 | 34.13% | 50.46% | 99.83% | **100.00%** | $3.11 \times 10^{12}$ | 14 | $1.22 \times 10^8$ | 198 |
| arb_rr | 8 | 86.32% | 92.64% | **100.00%** | **100.00%** | $3.11 \times 10^{12}$ | 10 | $1.20 \times 10^8$ | 1221 |
| bc_10 | 4 | 48.61% | 62.45% | **100.00%** | **100.00%** | $3.11 \times 10^{12}$ | 15 | $1.17 \times 10^8$ | 36 |
| cntr_gray | 256 | 65.56% | 82.16% | 99.40% | **100.00%** | $3.11 \times 10^{12}$ | 11 | $1.22 \times 10^8$ | 213 |
| fifoctl_s1_sf | 154 | 51.09% | 94.96% | **100.00%** | **100.00%** | $3.11 \times 10^{12}$ | 7 | $1.11 \times 10^8$ | 5742 |
| pulse_sync | 41 | 99.94% | 99.91% | **100.00%** | 99.99% | $3.11 \times 10^{12}$ | 8 | $1.15 \times 10^8$ | 18 |
| pulseack_sync | 148 | 93.97% | 97.50% | **100.00%** | **100.00%** | $3.11 \times 10^{12}$ | 8 | $1.16 \times 10^8$ | 3465 |
| sqrt_pipe | 2 | 78.72% | 25.43% | **100.00%** | **100.00%** | $3.11 \times 10^{12}$ | 3 | $1.07 \times 10^8$ | 372 |
| stackctl | 18 | 46.74% | 22.82% | **100.00%** | **100.00%** | $3.11 \times 10^{12}$ | 5 | $1.07 \times 10^8$ | 675 |
| Average | | 60.03% | 61.73% | 97.82% | **100.00%** | $3.12 \times 10^{12}$ | 5.00 | $1.10 \times 10^8$ | 414.00 |

**Baselines.** Our baseline methods are the state-of-the-art automated circuit design methods, including the BSD method Cheng et al. (2024a). Besides, we conduct a four-quadrant analysis comparing the design quality of our approach with that of three other models: our Deterministic HMM model, a state-of-the-art deterministic non-HMM model for automated circuit design(BSD), a state-of-the-art probabilistic HMM model for sequence modeling (LSTM), and a state-of-the-art probabilistic non-HMM model for sequence modeling (Transformer).

**Metrics.** For each circuit, we report the design quality with two metrics: the accuracy and size of the circuit design. The accuracy is calculated by simulation, with the average error rate of the circuit outputs with a set of sequential circuit inputs. The size is evaluated with the number of internal states and the number of logic gates using Synopsys Design Compiler.

More detailed experimental settings can be found in Appendix A.5.

## 5.1 COMPARE WITH THE STATE-OF-THE-ART

**Design Accuracy Comparison.** Table 1 shows the significant advantages of the proposed method on the VerilogEval v2 benchmark and DesignWare IP set in terms of circuit accuracy. The probabilistic non-HMM method (Transformer) and the Deterministic non-HMM method (BSD) both exhibit significantly lower accuracy. In contrast, methods incorporating HMM modeling, such as the probabilistic HMM model(LSTM) and our Deterministic HMM model Circuit HMM, achieve significantly higher accuracy of over 90% in most cases. Our Circuit HMM achieves 100% accuracy on all circuits except pulse_sync. The slight inaccuracy for pulse_sync arises from its single-output, multi-state nature, where it is challenging to infer state transition patterns from the $PO$ alone.

**Circuit Size Comparison.** Table 1 compares the circuit sizes in terms of the number of gates required, which shows that our proposed method CircuitHMM can obtain relatively compact circuit designs with fewer logic gates under near-perfect accuracy. The probabilistic non-HMM method

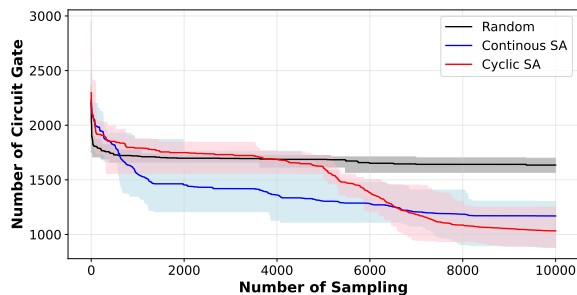

Figure 4: **State Encoding Searching Comparison.** Cyclic SA effectively reduces the circuit size with more sampling iterations.

(transformer) requires an impractical number of gates ($\sim 3.11 \times 10^{12}$) while still achieving very low accuracy. The probabilistic HMM method (LSTM) requires an equally impractical circuit size ($\sim 1.05 \times 10^{8}$ gates), although it achieves higher accuracy. The Deterministic non-HMM method (BSD), despite designing circuits with a very small number of gates, completely fails to meet the accuracy requirement of circuit design. As a result, its gate count is meaningless. In contrast, our Circuit HMM method achieves near-perfect accuracy ($> 99.99\%$) while ensuring the implementation of the circuits with minimal size ($\sim 400$) among practical approaches. See more details in Appendix A.6.

### 5.2 Evaluation of State Mining

To evaluate the capability of our state mining process in mining the reachable states of the target circuit, we compared it with the state-of-the-art deterministic sequence modeling methods. Although these methods cannot design sequential circuits end-to-end, they can find the inner states of the circuits effectively. Following the experimental settings of these works, we use AIGER's Brummayer et al. (2007) utility *aigfuzz* to generate 20 random automata and compare our state mining methods on both this randomly generated dataset and our automated circuit design dataset. On the randomly generated dataset, these methods (i.e., L* Steffen et al. (2012) and L# Tappler et al. (2019)) can only find $51.740\%$ and $45.035\%$ of the potential inner states found by our proposed method, and thus cannot design accurate circuits automatically without enough states. On the test dataset, these methods fail to find all reachable states in large circuits, with only 33.4% of reachable states found in Prob146, and a timeout for valid results in Prob155. More details are provided in Appendix A.7.

### 5.3 Effectiveness of State Encoding Optimization

Figure 4 shows their performance compared to random sampling on the 8-bit latch circuit under 10 trials. From the figure, it is evident that as the sampling continues, Continuous SA optimizes faster at the beginning, while Cyclic SA ultimately achieves the best encoding with 1033 gates compared to 1169 gates for Continuous SA (13% reduction) and 1635 gates (58% reduction). With fewer samplings (0-1000), all three methods can quickly identify better encoding. However, as the attempts continue, random sampling is inefficient due to the lack of a clear optimization direction. The Continuous SA with a continuously decreasing temperature, exhibits the fastest optimization speed in the early stages (0-2000), but becomes trapped in local optima due to the reduced exploration. The Cyclic SA method maintains a balance between exploration and optimization through periodic temperature adjustments. While its optimization speed is relatively slower within the initial 5000 iterations, Cyclic SA can rapidly optimize after exploration is nearly complete, ultimately achieving the optimal encoding. For detailed experimental result, see Appendix A.8.

### 6 Conclusion

The paper proposes Circuit HMM, a deterministic machine learning model for automated sequential circuit design. It formalizes the input-output relationships as a Markov Process, which reduces the complexity of the design space. With this computationally efficient model, we prove that the design accuracy of the sequential circuit converges to 100% with linear complexity. Experimental results demonstrate that the proposed method can accurately design real-world circuit modules comprising up to 5,000 logic gates, significantly outperforming the state-of-the-art. In 41 out of 43 cases, the design accuracy converges to 100% within 5 minutes.

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

# A APPENDIX

## A.1 A BRIEF EXAMPLE

Assume a sequential circuit with 1 inputs, 1 outputs, and 2 latch bits. The inputs are defined as $x$, the outputs as $y$, and the latch as $(l_0, l_1)$. At each time step, the outputs are computed as $y = (x \oplus l_0 \oplus l_1)$, where $\oplus$ denotes bitwise XOR. The latch state transition is as followed: $(l_0, l_1) = (l_0 \oplus x, l_0 \oplus l_1)$. We treat the circuit as a black-box circuit with inputs $x$ and outputs $y$ to demonstrate the complete flow of our method. Assume the initial state $s_0$ is $(l_0, l_1) = (0, 0)$, as shown in Figure 5(a). At time step $t = 0$, we prepare multiple random PI sequences of length 1 to sample the black-box circuit. For clarity, we list all possible PIs: $x = \{0, 1\}$. Based on the internal logic of the black-box circuit, we obtain the corresponding POs: $y = \{0, 1\}$. At this point, the two distinct POs correspond to two potential states. Since there are already different, we can observe that the states corresponding to PI sequences $[0]$ and $[1]$ are different states.

Next, we prepare a random PI sequence of length 2 to sample the black-box circuit under each potential state. For example, consider one possible PI sequence $[01]$, from which we obtain the PO sequences corresponding to the 2 potential states as $[01], [11]$, along with the PO sequence corresponding to $s_0$: $[01]$.

From the PO sequences, we observe that for the 2 potential states, the state corresponding to PI sequence $[0]$ matches the $s_0$, while the state corresponding to PI sequence $[1]$ is a new state $s_1$. As a result, we update the state diagram, as shown in Figure 5(b).

At time step $t = 1$, consider the same PI sample $x = \{0, 1\}$ for $s_1$ and $s_0$, we get PO respond as $y = \{0, 1, 1, 0\}$. Thus we find 4 potential states, represented as $PI$ sequence $[0][1][10][11]$. Next, we consider PI sequence $[01]$, and get corresponding $PO$ sequences $\{[01], [11], [00], [10]\}$. Since all $PO$ sequences are different, we consider states representing as $PI$ sequences $[0][1][10][11]$ as distinct states. Thus we obtain 2 new states $s_2, s_3$, as well as state transition function $s_0 \xrightarrow{x=0} s_0$ and $s_0 \xrightarrow{x=1} s_1$, also representing in $PI$ sequence format $[10], [11]$, as shown in Figure 5(c).

Next, we proceed to the $t = 2$ iteration. However, since the latch width is set to 2, we can find at most 4 distinct states. As a result, no new states will be discovered in subsequent iterations. But we can still obtain the state transition function $s_1 \xrightarrow{x=1} s_2$, $s_1 \xrightarrow{x=0} s_3$, $s_3 \xrightarrow{x=0} s_2$, $s_3 \xrightarrow{x=1} s_0$, $s_2 \xrightarrow{x=0} s_2$, $s_2 \xrightarrow{x=1} s_3$, as shown in Figure 5(d).

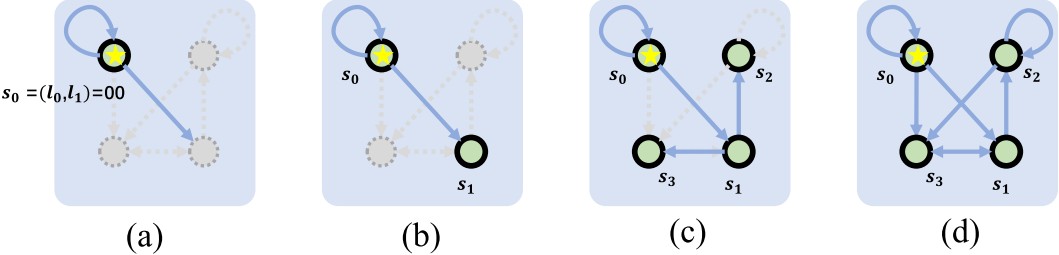

Figure 5: **An example of *State Mining* for a 4 states function.**

## A.2 THE STATE MINING ALGORITHM

We provide a pseudocode in algorithm 1 to further illustrate our state mining process for reproductivity.

---

**Algorithm 1** State Mining

---

**Require:** $r$ (initial state), $M$ (maximum iterations), $T$ (input sequence length), $N$ (sample count per state)
**Ensure:** $S$ (set of discovered states), $G$ (state transition graph), $P$ (input paths corresponding to states)
 1: Init $G \leftarrow \{\}, \quad Q \leftarrow [\,], \quad i \leftarrow 0$
 2: $Q.push(r), \quad S \leftarrow \{r\}, \quad P(r) \leftarrow [\,]$
 3: **while** $Q \neq \emptyset$ **and** $i < M$ **do**
 4: $\quad s \leftarrow Q.pop(), \quad i \leftarrow i + 1$
 5: $\quad$ **for** $j = 1$ **to** $N$ **do**
 6: $\qquad PI \leftarrow$ **GenerateRandSequence**$(length = 1)$
 7: $\qquad s_{next}, - \leftarrow$ **Sampling**$(s, PI)$
 8: $\qquad$ // *Given state and PI, generate the last state and PO*
 9: $\qquad s_{known} \leftarrow$ **FindSameState**$(S, s_{next})$
10: $\qquad$ **if** $s_{known}$ *is* $None$ **then**
11: $\qquad\quad Q.push(s_{next})$
12: $\qquad\quad P(s_{next}) \leftarrow Concat(P(s), PI)$
13: $\qquad\quad S \leftarrow S \cup \{s_{next}\}, \; G \leftarrow G \cup \{(s, PI, s_{next})\}$
14: $\qquad$ **else**
15: $\qquad\quad G \leftarrow G \cup \{(s, PI, s_{known})\}$
16: $\qquad$ **end if**
17: $\quad$ **end for**
18: **end while**
19: **return** $(S, G, P)$

20: Function **FindSameState**$(S, s)$:
21: **for** $st \in S$ **do**
22: $\quad PI \leftarrow$ **GenerateRandSequence**$(length = T)$
23: $\quad -, PO_s \leftarrow$ **Sampling**$(s, PI)$
24: $\quad -, PO_{st} \leftarrow$ **Sampling**$(st, PI)$
25: $\quad$ **if** $PO_s \equiv PO_{st}$ **then**
26: $\qquad$ **return** $st$
27: $\quad$ **end if**
28: **end for**
29: **return** $None$

---

### A.3 PROOF

#### A.3.1 CLAIM 1 AND ITS PROOF

**Claim 1** (Design Space Complexity Reduction). *Given a sequential circuit design problem, to precisely model a sequential circuit's complete behavior, the traditional temporal modeling design space complexity is $DS_{temp} \sim \mathcal{O}(2^{mt \cdot 2^{nt}})$, our FSM modeling design space complexity is $DS_{FSM} \sim \mathcal{O}(2^{(m+l) \cdot 2^{n+l}})$, and for sequential circuits with at least one PI ($n \geq 1$ and $l \geq 1$), FSM modeling achieves a double-exponential reduction in design space complexity $\sim \mathcal{O}(2^{m \cdot 2^{n \cdot 2^l}})$ compared to temporal modeling.*

Note that the observation period must span at least $t \geq 2^{w_{LI}}$ clock cycles to precisely model a sequential circuit's complete behavior.

*Proof.* For the complexity of the temporal design space, there are $2^{nt}$ possible input sequences over $t$ cycles, each requiring a mapping to one of $2^{mt}$ output sequences, yielding $2^{mt \cdot 2^{nt}}$ distinct behaviors. For the complexity of the FSM design space, the transition and output functions must define behavior for all $2^{n+l}$ state-input combinations, with each combination mapping to one of $2^l$ next states and one of $2^m$ outputs, resulting in $2^{(m+l) \cdot 2^{n+l}}$ possible implementations. Then we

consider the ratio of design space complexities:

$$\frac{|DS_{temp}|}{|DS_{FSM}|} = \frac{2^{mt \cdot 2^{nt}}}{2^{(m+l) \cdot 2^{n+l}}} = 2^{mt \cdot 2^{nt} - (m+l) \cdot 2^{n+l}} \tag{3}$$

For $t \geq 2^l$, we analyze the exponent $E$:

$$E = mt \cdot 2^{nt} - (m+l) \cdot 2^{n+l} \geq m \cdot 2^{l+n \cdot 2^l} - (m+l) \cdot 2^{n+l} \sim \mathcal{O}(m \cdot 2^{n \cdot 2^l}) \tag{4}$$

Therefore, the exponent $E$ grows double-exponentially with respect to $l$, making the ratio $\frac{|DS_{temp}|}{|DS_{FSM}|}$ triple-exponentially large ($\sim \mathcal{O}(2^{m \cdot 2^{n \cdot 2^l}})$). This demonstrates that FSM modeling achieves a triple-exponential reduction in design space complexity, as $|DS_{FSM}|$ is triple-exponentially smaller than $|DS_{temp}|$ for sequential circuits with $n \geq 1$ and $l \geq 1$. □

### A.3.2 PROOF OF LEMMA 1

*Proof.* Define the random variable

$$X_s = \begin{cases} 1, & \text{if state } s \text{ does not appear in the first } N \text{ samples,} \\ 0, & \text{otherwise.} \end{cases}$$

Then

$$\sum_{s \notin S_N} p(s) = \sum_{s \in S^*} p(s) X_s.$$

By the linearity of expectation:

$$\mathbb{E}[\sum_{s \notin S_N} p(s)] = \sum_{s \in S^*} p(s) \mathbb{E}[X_s].$$

For each state $s$, the probability of being sampled is $p(s)$. The probability of $s$ not appearing in $N$ independent samples is $(1 - p(s))^N$, so

$$\mathbb{E}[X_s] = (1 - p(s))^N.$$

Thus,

$$\mathbb{E}[\sum_{s \notin S_N} p(s)] = \sum_{s \in S^*} p(s)(1 - p(s))^N.$$

Note that for any $p \in [0, 1]$,

$$p(s)(1 - p(s))^N \leq \frac{1}{N+1},$$

and by treating the discrete sum as an upper bound weighted by a Dirac distribution, we obtain:

$$\sum_{s \in S^*} p(s)(1 - p(s))^N \leq \frac{\sum_s p(s)}{N+1}.$$

Because the probability distribution satisfies: $\sum_s p(s) = 1$

Therefore, we have:

$$\mathbb{E}[\sum_{s \notin S_N} p(s)] \leq \frac{1}{N+1} < \frac{1}{N}.$$

□

Table 2: The DesignWare IP information

| Circuit | PI | PO | Golden # states | Description |
|---|---|---|---|---|
| bictr_scnto | 11 | 9 | 256 | Up/Down Binary Counter with Static Count-to Flag |
| lfsr_load | 10 | 8 | 256 | LFSR Counter with Loadable Input |
| lfsr_scnto | 10 | 9 | 256 | LFSR Counter with Static Count-to Flag |
| lfsr_updn | 2 | 9 | 255 | LFSR Up/Down Counter |
| reg_s_pl | 10 | 8 | 256 | Register with Synchronous Enable Reset |
| shftreg | 7 | 4 | 16 | Shift Register |
| updn_ctr | 11 | 9 | 256 | Up/Down Counter |
| arb_rr | 10 | 7 | 8 | Arbiter with Round Robin Priority Scheme |
| bc_10 | 9 | 2 | 4 | Boundary Scan Cell Type BC_10 |
| cntr_gray | 11 | 8 | 256 | Gray Code Counter |
| fifoctl_s1_sf | 4 | 11 | 154 | Synchronous (Single-Clock) FIFO Controller with Static Flags |
| pulse_sync | 8 | 1 | 53 | Dual Clock Pulse Synchronizer |
| pulseack_sync | 8 | 3 | 148 | Pulse Synchronizer with Acknowledge |
| sqrt_pipe | 3 | 1 | 2 | Stallable Pipelined Square Root |
| stackctl | 2 | 10 | 18 | Synchronous (Single Clock) Stack Controller |

### A.3.3 PROOF OF THEOREM 2

*Proof.* According to lemma 1, for any state set $S^*$, the probability of undiscovered states $p(s)$ after $N$ samples satisfies:

$$\mathbb{E}[\sum_{s \notin S_N} p(s)] \leq \frac{1}{N}. \tag{5}$$

where $S_N$ denotes the set of states that are discovered after $N$ samples. The error rate $E(N)$ can be defined as

$$E(N) = \sum_{s \in S^*} p(s)e(s), \tag{6}$$

where $e(s)$ is the error rate of state $s$. $E(N)$ can be divided into 2 parts: when state $s$ is found, $e(s)$ can be reduced to 0 according to the BSD property; when state $s$ is not found, $e(s)$ can be considered as 1 since the learned model $C$ makes arbitrary errors on unfound states. So

$$E(N) = \sum_{s \notin S_N} p(s) \tag{7}$$

using the conclusion of equation 5, we have

$$\mathbb{E}[E(N)] = \mathbb{E}[\sum_{s \notin S_N} p(s)] \leq \frac{1}{N} \leq \epsilon. \tag{8}$$

$\square$

### A.4 SEQUENCE LENGTH ESTIMATION FOR STATE VALIDATION

**Proof of Theorem 1** Denote $R$ as the error rate of the miss validation of the circuit internal state $S_i$ and $S_j$ ($R \leftrightarrow S_i \neq S_j | PO_{seq}(S_i) = PO_{seq}(S_j)$), and $r$ as the average proportion of incorrect results of the circuit internal state. Since $PO_{seq}(S_i) = PO_{seq}(S_j)$ means that $PO$ is the same in every time step, the distribution of $R$ is

$$Pr(R) = \begin{cases} (1-r)^T & R = r \\ 1 - (1-r)^T & R = 0 \end{cases}. \tag{9}$$

So the expected value of $R$ is

$$E(R) = r(1-r)^T. \tag{10}$$

Considering the range of $r$ is $(0, \frac{1}{2}]$, $E(R)$ increases monotonically at $r \in (0, \frac{1}{T+1})$, and decreases monotonically at $r \in (\frac{1}{T+1}, \frac{1}{2}]$. Therefore, $E(R)$ takes the maximum value $\frac{1}{T+1} \left( \frac{T}{T+1} \right)^T$ when $r$

takes $\frac{1}{T+1}$, so

$$E(R) \leq \frac{1}{T+1}\left(\frac{T}{T+1}\right)^T < \frac{1}{T}. \tag{11}$$

Given a small value $\delta$, using Markov's inequality, we can obtain

$$Pr(R \geq \delta) \leq \frac{E(R)}{\delta} < \frac{1}{T\delta}. \tag{12}$$

Equation.12 means that in state validation, the probability that the error rate after determining $PO_{seq}(S_i) = PO_{seq}(S_j)$ is larger than $\delta$ is less than $\frac{1}{T\delta}$. Then let $T > k\frac{1}{\epsilon\delta}, k$ is a constant integer, such that the expectation of the design error $E(S_i \not\equiv S_j) < \epsilon$. $\qquad\square$

With the estimated transition matrix of the known states, the minimum required sequence length $T$ for a given $M_{seq}$ can be computed. Specifically, for two distinct states $S_p$ and $S_k$, we need to ensure that within $T$ steps, these states transition to two different states, $S'_p$ and $S'_k$, such that applying the same $PI$ generates different $PO$. Since the circuit output behavior and transition probabilities of unknown states are inaccessible, we estimate their behavior based on the properties of known states. Our goal is to ensure that within $T$ steps, the known state set contains states that can distinguish $S_p$ and $S_k$. Formally, for any known state $S^i$, the sequence length $T$ must satisfy:

$$P(S^i|S, T) > \tau \tag{13}$$

where $P(S^i|S, T)$ represents the probability of transitioning from state $S$ to state $S^i$ within $T$ steps, and $\tau$ is a predefined probability threshold.

Given the theorem 1, the targeted sequence length $T$, satisfying $\forall S_i \in \{S\}_{known}, P(\exists k < T, S_{t_k} = S^i|S_{t_0} = S^0) \geq \tau \approx 1$. In the next step, we need to estimate $T$ over the limited known states. First, let the set of states over $T$ steps be $S_{t_0}, S_{t_1}, \ldots, S_{t_{T-1}}$. Then:

$$P(\exists k < T, S_{t_k} = S^i|S_{t_0} = S^0) > \tau \tag{14}$$

$$\iff P(S^i \in \{S_{t_0}, S_{t_1}, \ldots, S_{t_{T-1}}\}) > \tau \tag{15}$$

$$\iff P(S^i \notin \{S_{t_0}, S_{t_1}, \ldots, S_{t_{T-1}}\}) \leq 1 - \tau \tag{16}$$

$$\iff \forall t < T, P(S_t \neq S^i) \leq 1 - \tau \tag{17}$$

$$\iff \forall t < T, P(P(S_t = S^i) = 0) \leq 1 - \tau \tag{18}$$

Second, we denote the state transition matrix as $A_{nn} = \{a_{ij}\}_{nn}$, where $n$ is the number of known states. We use random sampling to statistically estimate the transition frequencies of existing states to approximate the transition matrix. Then let $p_t^i = P(S_t = S^i)$.

$$\forall t < T, P(S_t = S^i) = 0 \tag{19}$$

$$\iff \forall t < T, p_t^i = 0 \tag{20}$$

$$\iff \forall t < T, \sum_{j=0}^{n-1} a_{ij}p_{t-1}^j = 0 \tag{21}$$

$$\impliedby \forall j, a_{ij} = 0 \tag{22}$$

In this case:

$$\forall t < T, P(p_t^i = 0) = P(S_t \neq S^i) \tag{23}$$

$$= \sum_{j=0,j\neq i}^{n-1} p_t^j < 1 - \tau \tag{24}$$

Since the $a_{i:} = 0$, as $T$ increases, $P(p_t^i = 0)$ decreases. Thus, the minimum $T$ satisfying the condition 24 will be found with the sequence length increasing.

Table 3: Detailed circuits size(# gates) comparison

| Circuits | Circuit Size(# gates) | | | | |
|---|---|---|---|---|---|
| | Prob | Prob+HMM | Deterministic | LLM | Ours |
| Prob041 | $3.12 \times 10^{12}$ | $1.17 \times 10^8$ | 8 | 24 | 48 |
| Prob046 | $3.12 \times 10^{12}$ | $1.17 \times 10^8$ | 8 | 24 | 48 |
| Prob047 | $3.12 \times 10^{12}$ | $1.17 \times 10^8$ | 8 | 24 | 48 |
| Prob067 | $3.11 \times 10^{12}$ | $1.04 \times 10^8$ | 1 | 196 | 60 |
| Prob085 | $3.11 \times 10^{12}$ | $1.12 \times 10^8$ | 9 | 36 | 78 |
| Prob088 | $3.11 \times 10^{12}$ | $1.03 \times 10^8$ | 1 | 63 | 6 |
| Prob089 | $3.11 \times 10^{12}$ | $1.03 \times 10^8$ | 1 | 56 | 24 |
| Prob096 | $3.11 \times 10^{12}$ | $1.03 \times 10^8$ | 1 | 197 | 33 |
| Prob107 | $3.11 \times 10^{12}$ | $1.03 \times 10^8$ | 1 | 74 | 6 |
| Prob109 | $3.11 \times 10^{12}$ | $1.03 \times 10^8$ | 1 | 68 | 6 |
| Prob110 | $3.11 \times 10^{12}$ | $1.05 \times 10^8$ | 2 | 24 | 12 |
| Prob111 | $3.11 \times 10^{12}$ | $1.05 \times 10^8$ | 2 | 24 | 12 |
| Prob119 | $3.11 \times 10^{12}$ | $1.03 \times 10^8$ | 1 | 68 | 18 |
| Prob120 | $3.11 \times 10^{12}$ | $1.03 \times 10^8$ | 1 | 120 | 18 |
| Prob121 | $3.11 \times 10^{12}$ | $1.03 \times 10^8$ | 1 | 146 | 36 |
| Prob127 | $3.11 \times 10^{12}$ | $1.05 \times 10^8$ | 2 | 78 | 15 |
| Prob128 | $3.11 \times 10^{12}$ | $1.15 \times 10^8$ | 8 | 69 | 15 |
| Prob133 | $3.11 \times 10^{12}$ | $1.05 \times 10^8$ | 2 | 306 | 54 |
| Prob136 | $3.11 \times 10^{12}$ | $1.03 \times 10^8$ | 1 | 155 | 33 |
| Prob137 | $3.11 \times 10^{12}$ | $1.03 \times 10^8$ | 1 | 402 | 57 |
| Prob138 | $3.11 \times 10^{12}$ | $1.03 \times 10^8$ | 1 | 151 | 39 |
| Prob140 | $3.11 \times 10^{12}$ | $1.04 \times 10^8$ | 1 | 635 | 81 |
| Prob142 | $3.11 \times 10^{12}$ | $1.07 \times 10^8$ | 3 | 189 | 30 |
| Prob146 | $3.13 \times 10^{12}$ | $1.05 \times 10^8$ | 1 | 1605 | 2181 |
| Prob149 | $3.11 \times 10^{12}$ | $1.07 \times 10^8$ | 3 | 935 | 96 |
| Prob152 | $3.11 \times 10^{12}$ | $1.09 \times 10^8$ | 4 | 265 | 81 |
| Prob155 | $3.11 \times 10^{12}$ | $1.09 \times 10^8$ | 4 | 813 | 357 |
| bictr_scnto | $3.12 \times 10^{12}$ | $1.22 \times 10^8$ | 11 | - | 189 |
| lfsr_load | $3.12 \times 10^{12}$ | $1.20 \times 10^8$ | 10 | - | 597 |
| lfsr_scnto | $3.12 \times 10^{12}$ | $1.20 \times 10^8$ | 10 | - | 129 |
| lfsr_updn | $3.11 \times 10^{12}$ | $1.07 \times 10^8$ | 2 | - | 879 |
| reg_s_pl | $3.11 \times 10^{12}$ | $1.20 \times 10^8$ | 10 | - | 96 |
| shftreg | $3.11 \times 10^{12}$ | $1.14 \times 10^8$ | 7 | - | 66 |
| updn_ctr | $3.11 \times 10^{12}$ | $1.22 \times 10^8$ | 14 | - | 198 |
| arb_rr | $3.11 \times 10^{12}$ | $1.20 \times 10^8$ | 10 | - | 1221 |
| bc_10 | $3.11 \times 10^{12}$ | $1.17 \times 10^8$ | 15 | - | 36 |
| cntr_gray | $3.11 \times 10^{12}$ | $1.22 \times 10^8$ | 11 | - | 213 |
| fifoctl_s1_sf | $3.11 \times 10^{12}$ | $1.11 \times 10^8$ | 7 | - | 5742 |
| pulse_sync | $3.11 \times 10^{12}$ | $1.15 \times 10^8$ | 8 | - | 18 |
| pulseack_sync | $3.11 \times 10^{12}$ | $1.16 \times 10^8$ | 8 | - | 3465 |
| sqrt_pipe | $3.11 \times 10^{12}$ | $1.07 \times 10^8$ | 3 | - | 372 |
| stackctl | $3.11 \times 10^{12}$ | $1.07 \times 10^8$ | 5 | - | 675 |

Furthermore, $\tau$ can be determined by the number of sequences $M_{seq}$. We only need to ensure that at least one distinguishing sequence exists among $M_{seq}$ sequences, i.e.,

$$M_{seq} * P(\exists k < T, S_{t_k} = S^i | S_{t_0} = S^0) > M_{seq} * \tau \geq 1 \tag{25}$$

$$P(\exists k < T, S_{t_k} = S^i | S_{t_0} = S^0) \geq \frac{1}{M_{seq}} \tag{26}$$

Using the transition matrix of the known states, the minimum required sequence length $T$ for a given $M_{seq}$ can be computed.

## A.5 EXPERIMENTAL SETUP

### A.5.1 DATASETS

**VerilogEval v2.** VerilogEval V2 is a specialized benchmarking framework designed to evaluate the performance of methods in generating Verilog code for hardware design tasks. It leverages a dataset sourced from HDLBits, a collection of digital circuit design exercises, featuring 156 carefully curated problems. These problems span a wide range of Verilog coding tasks, including the implementation of simple combinational circuits, complex finite state machines, code debugging, and testbench construction. The framework focuses on generating self-contained Verilog modules, ensuring that the generated code does not rely on the instantiation of other modules. Functional correctness is evaluated through automated testing by comparing the simulation outputs of generated code against golden reference solutions. Our experiment selected 22 sequential circuits from the VerilogEval v2 dataset. The selected 22 sequential circuits have internal states ranging from 1 to over 2000, capable of evaluating the generation quality on circuits with varying levels of complexity.

**DesignWare IP.** The DesignWare Library's Datapath and Building Block IP is a collection of reusable blocks. The large availability of IP components enables design reuse and significantly improves productivity. To further demonstrate the practicality and versatility of our method, we extend the evaluation to a dataset selected from the DesignWare library. This dataset includes a diverse set of real-world sequential circuits, covering categories such as control logic, arithmetic components and registers. Detailed information on our selected circuits are shown in Table 2.

### A.5.2 BASELINES

**Transformer.** For non-HMM probabilistic models, we use a Transformer with a 2-layer encoder-decoder architecture, 2 attention heads, and a hidden dimension of 64, where $PI$ and $PO$ are treated as tokens, and the model generates $PO$ tokens autoregressively. Each $PO$ bit is set to 1 if its prediction exceeds $0.5$.

**LSTM.** For probabilistic HMM models, we adopt an LSTM with probabilistic state representation. The LSTM configuration uses 2 layers and 20 hidden features, with input_dim and output_dim set to the $PI$ and $PO$ widths, respectively. Each $PO$ bit is set to 1 if its prediction exceeds $0.5$.

**BSD.** For deterministic non-HMM models, we use BSD, which provides a deterministic baseline without HMM modeling. We set the sample amount to 40000 and it always samples from the circuit that is associated with the initial state.

### A.5.3 METRICS

For each circuit, we report the design quality with two metrics, i.e., the accuracy of the output sequences, calculated as the average correctness rate of PO, and the size of the generated sequential circuits. To determine the number of gates of the generated circuits for the Transformer and LSTM, we count the floating-point multiplications required during inference and multiply them by the number of gates needed for a single floating-point multiplication (based on the XiangShan CPU implementation Xu et al. (2022)).

To ensure a fair and quantitative comparison, the training set for the end-to-end NNs (Transformer, LSTM) consists of a large, fixed dataset of 100,000 IO examples, each a 100-length sequence. And our proposed method, in its state mining phase, uses an average of 500 IO examples per discovered state.

Since the state number is unknown, the $(PI, PO)$ sequence length is fixed to 100 in training and 500 in inference for in Transformer and LSTM. For Circuit HMM, the sequence length can be more flexible on-demand in training with the help of HMM evaluation. We evaluate the Acc. of our proposed Circuit HMM using 10000 randomly generated sequences, each of length 10000. It is worth noting that the evaluation of our proposed Circuit HMM is conducted under stricter conditions compared to the baseline methods, as the sequence length for testing is significantly longer. This stricter evaluation setup makes it easier to identify potential design errors, which demonstrates its robustness and reliability in handling complex and extensive input-output relationships.

## A.6 DETAIL OF CIRCUIT SIZE COMPARISON

We show the detailed results of the size of the designed circuits in table 3. The results demonstrate that the probabilistic method and probabilistic HMM methods produce circuits that are orders of magnitude larger than those generated by our Circuit HMM method. While the deterministic method(BSD) achieves smaller circuit sizes, its limited ability to handle diverse input-output patterns often compromises the functional correctness of the generated circuits. Due to its inability to handle the diverse input-output patterns of sequential circuits, it often generates meaningless circuits, resulting in accuracy below 50% for some cases. In contrast, our Circuit HMM method achieves both practical circuit sizes and near-perfect accuracy. For instance, the average circuit size is 129.33 gates for VerilogEval circuits and 926.40 gates for the DW IP set, representing reductions of 5–7 orders of magnitude compared to the probabilistic HMM method.

Specifically, the SOTA circuit design method (the deterministic method: BSD Cheng et al. (2024a)) only achieves 60% accuracy on a circuit with 36 logic gates. Therefore, compared to SOTA, we have increased the scale by two orders of magnitude in accurately generating sequential circuits, pushing the scale limit of the automated logic design. In fact, circuits with fewer than 5000 logic gates are practical in the real world (a 32-bit full adder contains only approximately 160 logic gates, furthermore, our DW benchmark can cover most of the commonly used IPs), demonstrating that the proposed method is applicable.

Although the VerilogEval v2 dataset provides additional inputs for evaluating LLM-generated circuits, it is important to note that these methods operate under different assumptions and input formats compared to the baseline methods presented here. As such, direct comparisons between LLM methods and our Circuit HMM method are not entirely fair. Consequently, the results for LLM-generated circuits are provided as a reference only and are not included in the detailed analysis of baseline methods.

## A.7 EVALUATION OF STATE MINING

Table 5 shows the performance of our state mining and the existing sequence modeling methods Tappler et al. (2019); Steffen et al. (2012). The input bit width of the generated circuits ranges from 6 to 18, and the output bit width ranges from 2 to 50. For the automata learning methods compared with our approach, we use the automata learning library proposed in Muškardin et al. (2022). If the algorithm fails to obtain a feasible solution, the number of discovered states is recorded as 1, representing the given initial state.

We also make further comparison in table 4. In our design process, the accuracy of not using state mining is 60.03% (Section 5.1), the average accuracy of using the SOTA method ( L*& L#) for state mining is 34.89% & 34.37% (table 4, we only show 10/42 circuits where SOTA methods fail to find the correct states), and the average accuracy of using our method for state mining is 100.00%. This not only demonstrates the effectiveness of our algorithm (as detailed in Appendix A.7 for data comparison), but also shows that our design flow is necessary and effective for addressing the critical issue of insufficient accuracy in the automatic design of sequential circuits.

As shown in the results, our method can explore more states than the automata learning methods in almost all circuits. This is because our state validation process effectively distinguishes newly discovered states from existing ones, minimizing the merging of states that exhibit similar behavior on the *PO*.

## A.8 EVALUATION OF STATE ENCODING

We have evaluated the SA-based encoding optimization on a subset of the benchmarks, and the results shown in table 6 consistently show that it reduces circuit area by approximately 85.54% on average compared to an intuitive direct encoding. The results demonstrate the critical role of our state encoding optimization in minimizing circuit size.

Table 4: Accuracy Comparison of State Mining with SOTA

| circuit_name | L* | L# | Ours |
|---|---|---|---|
| Prob121 | 80.00% | 80.00% | 100.00% |
| Prob146 | 33.39% | 33.39% | 100.00% |
| Prob155 | 25.00% | 20.45% | 100.00% |
| bictr_scnto | T/o | T/o | 100.00% |
| lfsr_updn | 96.47% | 95.29% | 100.00% |
| updn_ctr | T/o | T/o | 100.00% |
| arb_rr | 25.00% | 25.00% | 100.00% |
| cntr_gray | T/o | T/o | 100.00% |
| pulse_sync | 20.75% | 22.64% | 99.99% |
| pulseack_sync | 68.24% | 66.89% | 100.00% |
| Average | 34.89% | 34.37% | 100.00% |

*Note: T/o indicates timeout (exceeded 6-hour time limit).*

Table 5: Circuit information and state found.

| Circuit name | Circuit info | | State found | | |
|---|---|---|---|---|---|
| | Input | Output | L* | L# | Ours |
| random00 | 9 | 10 | 7 | 6 | 22 |
| random01 | 6 | 7 | 1 | 1 | 3 |
| random02 | 11 | 15 | 24 | 14 | 48 |
| random03 | 8 | 2 | 1 | 1 | 4 |
| random04 | 10 | 3 | 1 | 1 | 2 |
| random05 | 18 | 21 | 1 | 1 | 5 |
| random06 | 16 | 11 | 1 | 1 | 16 |
| random07 | 8 | 50 | 25 | 25 | 25 |
| random08 | 7 | 10 | 1 | 1 | 3 |
| random09 | 8 | 6 | 2 | 1 | 2 |
| random10 | 8 | 14 | 31 | 24 | 31 |
| random11 | 6 | 16 | 1 | 1 | 3 |
| random12 | 14 | 3 | 1 | 1 | 4 |
| random13 | 6 | 11 | 2 | 2 | 3 |
| random14 | 6 | 24 | 5 | 5 | 5 |
| random15 | 6 | 2 | 1 | 1 | 3 |
| random16 | 16 | 2 | 1 | 1 | 3 |
| random17 | 9 | 43 | 4 | 4 | 4 |
| random18 | 6 | 33 | 12 | 12 | 12 |
| random19 | 8 | 13 | 47 | 40 | 49 |
| Geomean | 8.70 | 9.72 | 3.08 | 2.81 | 6.73 |

## A.9 RUNTIME EFFICIENCY COMPARISON

Regarding runtime efficiency, LSTM and Circuit HMM, with a complexity of $O(1)$, significantly outperform transformers with a complexity of $O(T^3)$. Due to the Markov properties of LSTM and Circuit HMM, each output at runtime depends on only implicit or explicit states and PI at this time, resulting in a runtime complexity proportional to the circuit size, which is $O(1)$. Further, with the refined logic function representation, the small number of circuit gates of Circuit HMM ensures its

Table 6: Comparison on Circuit Size Optimize Methods

| Circuit Name | Random | Ours | Reduction Rate |
|---|---|---|---|
| lfsr_updn | 2637 | 879 | 66.67% |
| reg_s_pl | 2100 | 96 | 95.43% |
| shftreg | 144 | 66 | 54.17% |
| updn_ctr | 3687 | 198 | 94.63% |
| Average | 2142 | 309.75 | 85.54% |

efficient execution. For the Transformer lacks Markov properties, its current output depends on all previous PI sequences. On the one hand, the attention operation introduces $O(T^2)$ complexity. On the other hand, as the new PI must concatenate previous PI sequences for input to the encoder at each time step and followed by autoregressive generation of all PO sequences from time 0 in the decoder, each inference introduces another $O(T)$ complexity. Thus, the overall runtime complexity of $O(T^3)$ makes transformers infeasible to serve as a sequential circuit with very long sequences.

### A.10 BROADER IMPACT

Our proposed method enhances the automation of sequential circuit design, reducing the reliance on manual design efforts and lowering the associated development costs. Furthermore, it can be integrated into existing EDA tools, providing scalable solutions for diverse application-specific circuits in the IoT era.

### A.11 PRELIMINARY AND RELATED WORKS

**Logic optimization.** Logic optimization aims to optimize the area/timing/power performance. Roy et al. (2021) and Song et al. (2024) utilized deep neural network models to design prefix circuits, which are commonly used in arithmetic operations. Additionally, Lin et al. (2024) proposed a divide-and-conquer method for circuit design, while Zuo et al. (2023) introduced a reinforcement learning (RL)-based approach. These methods leverage advanced machine learning techniques to address the unique challenges of specific circuit design tasks.

**Binary Speculation Diagram.** The Binary Speculation Diagram (BSD) Cheng et al. (2024a) is a novel probability graph model for large-scale circuit design from input-output examples. As an approximation version of the Binary Decision Diagram(BDD), BSD expands in a given bit sequence, and gradually builds a rooted, directed acyclic graph (DAG) which consists of internal decision nodes and leaf speculation nodes. The internal decision node represents a Boolean variable, assigning a value of 0 or 1 to its two child nodes, while the speculation nodes approximate the sub-functions of the child nodes by treating them as constant 0 or 1.

During the expansion process, BSD iteratively approximates functional equivalence to minimize the number of newly introduced BSD nodes. Functional equivalence is evaluated using a Monte Carlo method, which relies on the same set of expanded variables by sampling a large set of inputs and verifying if the outputs match. After each iteration, the speculative node set and the BSD are updated, with the expanded nodes and their new child nodes treated as speculative nodes for further expansion. This iterative process continues until the BSD successfully passes verification. To get a more efficient BSD simplification, BSD also employs a novel cluster approach based on a Boolean Distance. By iteratively expanding the BSD guided by Boolean Distance, the final BSD with ultra-high accuracy can be generated efficiently.

**IOHMM.** The Input-Output Hidden Markov Model (IOHMM) is an extension of the traditional Hidden Markov Model (HMM) that incorporates external input variables into its framework. While a standard HMM models a sequence of observed states as being generated by a hidden Markov process, the IOHMM allows the transition probabilities between hidden states and the emission probabilities of observations to be conditioned on external inputs. This makes IOHMMs particularly suitable for modeling time-series data where external factors influence the dynamics of the system.

Formally, an IOHMM consists of three main components: (1) **Hidden States:** A finite set of unobserved (latent) states that evolve according to a Markov process. (2) **Input Variables:** External variables that influence the transition between hidden states and the generation of observations. (3) **Output Variables:** Observed data that are emitted based on the hidden state and possibly conditioned on the input variables.

The key difference between an HMM and an IOHMM lies in the parameterization of the transition and emission probabilities. In an IOHMM, these probabilities are often modeled using functions, such as neural networks or logistic regression, that take the input variables into account. This allows the IOHMM to capture more complex dependencies between the inputs and the sequence dynamics compared to a standard HMM.

IOHMMs have been applied in various domains, including speech recognition, financial modeling, and biological sequence analysis, where external inputs play a crucial role in influencing the underlying processes. Their ability to integrate external information with sequential data makes them a powerful tool for predictive modeling in dynamic systems.

**Transformer.** The Transformer Vaswani (2017) represents a paradigm shift in the field of artificial intelligence, particularly in natural language processing (NLP). Unlike traditional sequence-to-sequence models that rely heavily on recurrent or convolutional architectures, the Transformer is based entirely on the self-attention mechanism, enabling it to process sequences in parallel and capture long-range dependencies efficiently.

At the core of the Transformer is the self-attention mechanism, which computes a weighted representation of input tokens by attending to all other tokens in the sequence. This mechanism is complemented by positional encodings, which provide the model with information about the order of tokens, addressing the lack of inherent sequential structure in its architecture.

The architecture consists of an encoder-decoder structure, where both the encoder and decoder are composed of stacked layers of multi-head self-attention and feed-forward networks. The encoder processes input sequences into a continuous representation, while the decoder generates output sequences by attending to both the encoder outputs and previously generated tokens.

## A.12 THE USE OF LLMS

During the writing of this paper, we used LLMs for grammar checking and correction. The ideation of this research relies on the authors' intellectual contributions, with no involvement of LLMs. The use of LLMs in the experiment for detailed circuit size comparison is already described in Section A.6.

