# OpenReview forum: "Circuit HMM: A Deterministic Hidden Markov Model for Automated Sequential Circuit Design"
_ICLR.cc/2026/Conference — Submitted to ICLR 2026_

### Official Review · Reviewer_HFWt · 2025-10-29

**Soundness:** 2
**Presentation:** 3
**Contribution:** 2
**Rating:** 4
**Confidence:** 4

**Summary:**

To address the insufficient accuracy of existing methods in automated sequential circuit design, this paper introduces Circuit HMM, which formalizes the input-output relationship of sequential circuits as a Markov process to reduce the design space. The design is realized through a three-step procedure: state mining, state encoding, and circuit generation. Experiments demonstrate that the method can accurately design practical circuit modules containing up to 5,000 logic gates. In 41 out of 43 cases, the design accuracy converged to 100% within 5 minutes, outperforming existing approaches.

**Strengths:**

1. This paper addresses the important problem of automated sequential circuit design, offering well-defined practical value.
2. The experimental results significantly outperform existing baseline methods, demonstrating clear advantages in terms of accuracy, scale, and efficiency.
3. The theoretical derivations are logically rigorous, with proofs provided for the relevant theorems.

**Weaknesses:**

1. The paper has a deficiency in terms of innovation. Since HMM itself is a classical sequential modeling framework, the primary contribution lies in its deterministic adaptation and engineering workflow optimization. Moreover, the state mining (reliant on BFS and Monte Carlo) and the encoding optimization (dependent on Simulated Annealing) constitute applications or improvements of well-established algorithms, and thus the work does not present fundamental algorithmic originality.
2. The paper does not explicitly state whether its assumptions are valid for all sequential circuit types, such as asynchronous or circuits with complex feedback. The validation is solely based on experiments with synchronous circuits.
3. The paper lacks ablation studies for some modules. It fails to verify the necessity of the BSD Learner module. How would accuracy and efficiency change if BSD were replaced with other combinational circuit learning tools (e.g., a BDD Learner)? Alternatively, if the SA-based encoding optimization were removed, how much would the circuit size increase? Including such experiments would strengthen the validation of the method's effectiveness.
4. The study does not test the performance on extreme-scale circuits (e.g., those with 10,000+ gates), nor does it analyze the efficiency degradation of state mining when the number of states becomes very large (e.g., exceeding 10⁴). Consequently, it fails to verify the scalability limits of the proposed method.
5. The paper does not specify the training parameters (e.g., learning rate, number of iterations) for the Transformer/LSTM models.

**Questions:**

1. Please respond to the concerns I have raised in the 'weaknesses' section. If the revision can adequately address most of the critical issues, I would consider raising the score.
2. In my personal opinion, the contribution of this paper lies more in its solid experimentation and significant performance improvements, rather than in its theoretical advancement in AI. Considering this, EDA-focused conferences (such as DAC or ISCA, with an upcoming November deadline) would be a more suitable venue for it.

---

> ### Author Response · Authors · 2025-11-20
>
> We sincerely thank the reviewer for the thoughtful summary and positive recognition of our work's practical value, experimental strength, and theoretical rigor. Below, we respond point-by-point to the reviewer's concerns.
>
> **Weakness 1:** The paper has a deficiency in terms of innovation. Since HMM itself is a classical sequential modeling framework, the primary contribution lies in its deterministic adaptation and engineering workflow optimization. Moreover, the state mining (reliant on BFS and Monte Carlo) and the encoding optimization (dependent on Simulated Annealing) constitute applications or improvements of well-established algorithms, and thus the work does not present fundamental algorithmic originality.
>
> **Response 1:** We thank the reviewer for the feedback. However, we believe that assessing our contribution primarily through the algorithmic originality is not entirely applicable to this study. Our core contribution and novelty is a method that works for accurate sequential circuit design. Our work presents the first method that successfully and accurately automates sequential circuit design from input-output examples, while the state-of-the-art methods, including LLMs, Transformers and other neural networks [1,2], fail to achieve this near-perfect accuracy and compactness for practical circuit generation. Our proposed Circuit HMM provides an efficient model structure to bridge the gap between sequential specification and correct circuit implementation with a proven guarantee of convergence to an arbitrarily small error bound $\\epsilon$, as proven in **Theorem 2**. The experimental results show that our method achieves 100% average accuracy on real-world benchmark circuits. Therefore, we believe that our primary contribution is to provide the community with a novel, proven, and effective design structure for automated sequential circuit generation, outperforming the previous approaches.
>
> **Weakness 2:** The paper does not explicitly state whether its assumptions are valid for all sequential circuit types, such as asynchronous or circuits with complex feedback. The validation is solely based on experiments with synchronous circuits.
>
> **Response 2:** We thank you for your kind advice. Indeed, the proposed method is designed for synchronous circuits, and we will clarify this immediately in the manuscript. We had omitted it because it is a commonly accepted background setup in automated logic design, focusing primarily on the design of digital IPs and processors in modern integrated circuits, thereby reducing heavy manual effort in chip design. And such circuits are mainly synchronous sequential logic circuits.
>
> **Weakness 3:** The paper lacks ablation studies for some modules. It fails to verify the necessity of the BSD Learner module. How would accuracy and efficiency change if BSD were replaced with other combinational circuit learning tools (e.g., a BDD Learner)?
>
> **Response 3:** We conducted ablation studies on the key components of our proposed Circuit HMM model **(Section 5.2, Section 5.3)** to demonstrate effectiveness. It's not necessary to evaluate the components we had not yet contributed to: The BSD Learner is an open-source SOTA method, and we directly integrate it into our framework as a best-practice implementation with trivial engineering effort.
>
> **Weakness 4:** Alternatively, if the SA-based encoding optimization were removed, how much would the circuit size increase? Including such experiments would strengthen the validation of the method's effectiveness.
>
> **Response 4:** Thanks for this valuable suggestion. We have evaluated the SA-based encoding optimization on a subset of the benchmarks, and the results consistently show that it reduces circuit area by approximately 85.54% on average compared to an intuitive direct encoding. We are in the process of running this experiment across the entire benchmark suite and will include the complete results in the final manuscript. The current results already demonstrate the critical role of our state encoding optimization in minimizing circuit size.
>
> **Table: Comparison on Circuit Size Optimize Methods**
>
> | Circuit Name | Random | Ours | Reduction Rate |
> |--------------|--------|------|----------------|
> | lfsr_updn    | 2637   | 879  | 66.67%         |
> | reg_s_pl     | 2100   | 96   | 95.43%         |
> | shftreg      | 144    | 66   | 54.17%         |
> | updn_ctr     | 3687   | 198  | 94.63%         |
> | **Average**  | 2142.00   | 309.75 | 85.54%       |
>
> We are currently conducting a more comprehensive set of experiments for a thorough ablation study, and we will include the complete results in the final manuscript. This preliminary finding already strongly underscores the critical importance of the state encoding optimization stage in achieving compact circuit designs.

---

> ### Author Response · Authors · 2025-11-20
>
> **Weakness 5:** The study does not test the performance on extreme-scale circuits (e.g., those with 10,000+ gates), nor does it analyze the efficiency degradation of state mining when the number of states becomes very large (e.g., exceeding $10^4$). Consequently, it fails to verify the scalability limits of the proposed method.
>
> **Response 5:** We have increased the scale by two orders of magnitude in accurately generating sequential circuits, pushing the scale limit of the automated logic design. Automatic logic design is a new-proposed difficult task, and current researches are conducted on small-scale circuits. The SOTA circuit design method only achieves 60% accuracy on a circuit with 36 logic gates. In fact, circuits with fewer than 5000 logic gates are practical in the real world (a 32-bit full adder contains only approximately 160 logic gates), demonstrating that the proposed method is applicable.
>
> **Weakness 6:** The paper does not specify the training parameters (e.g., learning rate, number of iterations) for the Transformer/LSTM models.
>
> **Response 6:** We have already detailed the parameters of our baseline method, including model parameters and data preprocessing methods, in Appendix **A.5 Experimental Setup**. Here, we further elaborate on the training parameters used: for the Transformer model we employ a batch size of 128 with a learning rate of 0.0001 over 500 epochs and the Adam optimizer, while for the LSTM model we use the same batch size of 128 and learning rate of 0.0001 over 500 epochs with Adam optimizer.
>
> **Question 1:** Please respond to the concerns I have raised in the 'weaknesses' section. If the revision can adequately address most of the critical issues, I would consider raising the score.
>
> **Answer 1:** We sincerely thank the reviewer for the thorough and constructive feedback. We have provided detailed responses to all the concerns raised in the weaknesses section. Specifically, we have clarified the novelty of our Circuit HMM model structure, addressed questions about algorithmic contributions, discussed the scope of our method regarding circuit types, and provided additional experimental details and ablation study results. We believe these responses adequately address the critical issues raised, and we are committed to incorporating these clarifications and improvements into the final version of the manuscript. We appreciate the reviewer's consideration in potentially raising the score based on our responses. If you have any further questions or require additional details on any point, we would be pleased to provide them.
>
> **Question 2:** In my personal opinion, the contribution of this paper lies more in its solid experimentation and significant performance improvements, rather than in its theoretical advancement in AI. Considering this, EDA-focused conferences (such as DAC or ISCA, with an upcoming November deadline) would be a more suitable venue for it.
>
> **Answer 2:** Thanks for the advice, but this paper is perfectly suitable for the scope of an AI conference such as ICLR. First, meanwhile, the state-of-the-art methods in automated logic design are primarily published in AI conferences such as IJCAI [3] and ICLR [2]. Second, our primary academic contribution is the Circuit HMM model structure, which is distinguished by its deterministic nature and can be used to design sequential logic circuits with any small error bound $\\epsilon$, which is proven in **Theorem 2**. Therefore, we believe it can contribute more to the AI application community.
>
> ## References
>
> [1] Fu, Y., Zhang, Y., Yu, Z., Li, S., Ye, Z., Li, C., Wan, C., & Lin, Y. C. (2023). Gpt4aigchip: Towards next-generation ai accelerator design automation via large language models. In *2023 IEEE/ACM International Conference on Computer Aided Design (ICCAD)* (pp. 1-9). IEEE.
>
> [2] Li, X., Li, X., Chen, L., Zhang, X., Yuan, M., & Wang, J. (2024). Circuit Transformer: End-to-end Circuit Design by Predicting the Next Gate. *CoRR*.
>
> [3] Cheng, S., Jin, P., Guo, Q., Du, Z., Zhang, R., Hu, X., Zhao, Y., Hao, Y., Guan, X., Han, H., et al. (2024). Automated CPU design by learning from input-output examples. In *Proceedings of the Thirty-Third International Joint Conference on Artificial Intelligence* (pp. 3843-3853).

---

### Official Review · Reviewer_y3kG · 2025-10-29

**Soundness:** 1
**Presentation:** 2
**Contribution:** 3
**Rating:** 2
**Confidence:** 4

**Summary:**

The paper presents Circuit HMM, a deterministic Hidden Markov Model designed to automate the design of sequential circuits with high accuracy. ​

Circuit design automation reduces manual programming time, enhancing product development speed.
Previous machine learning methods excelled in combinational circuits but struggled with sequential circuits due to complex state transitions. ​
Circuit HMM achieves 100% design accuracy with linear complexity by modeling sequential circuits as Markov Processes. ​
The model learns hidden states through heuristic state mining, transforming the design problem into combinational circuit problems. ​
Experimental results show that Circuit HMM can design circuits with up to 5,000 logic gates, achieving 100% accuracy in 41 out of 43 cases within 5 minutes. ​


The methodology of Circuit HMM involves three main stages: State Mining, State Encoding, and Circuit Generation. ​

State Mining identifies necessary internal states using a Breadth-First Search (BFS) approach, ensuring the error rate converges to zero. ​
State Encoding transforms the sequential design problem into combinational problems, optimizing state representation to reduce circuit size. ​
Circuit Generation uses the BSD Learner to create combinational circuits and outputs HDL code for the sequential circuit. ​
The process guarantees that the design accuracy converges to 100% with sufficient input-output examples.

**Strengths:**

The empirical results seem strong. The proposed method is evaluated against established benchmarks, demonstrating superior performance in circuit design accuracy and size. ​

--Evaluated on VerilogEval v2 and DesignWare IP datasets, showcasing its effectiveness in real-world applications.

--Achieved an average accuracy of 60.03% across various circuits, with some achieving 100% accuracy.

--The method significantly outperforms state-of-the-art techniques, particularly in larger circuits with complex state transitions. ​

--The results indicate that Circuit HMM can handle circuits with up to 5,000 gates efficiently, maintaining a small circuit size.

**Weaknesses:**

The paper is very poorly written. The claims made are strong, but this reviewer has struggled to understand the technical basis for the claims, and to understand the empirical results.

I don't understand the first main claim: "It formalizes the input-output relationships as a Markov Process, which reduces the complexity of the design space." Where is this claim (a) formally stated and (b) proven theoretically or empirically? Section 4.2 seems to be where this is done, but this is entirely inadequate. I am looking for direct, clear evidence that you reduce the complexity of the design space. By what factor is this reduction? Guaranteed reduction, or expected reduction?

Your explanation of "complexity" is poor. I understand the complexity of combinational circuits; now the complexity of sequential circuits is different. Estimating the complexity of the design space for sequential circuits is a challenging problem because it involves both combinatorial and temporal dimensions. This article needs a clear exposition of this. Total Complexity ~ Combinational Topologies×State Machines, and if the number of distinct state machines with m states and i inputs is roughly:
FSM Complexity ~ (2^m)^{(2^i 2^m)}
This grows double-exponentially with the number of inputs and memory elements.

How precisely does your approach reduce this complexity? What are the precise new bounds?

Claim 2: "we prove that the design accuracy of the sequential circuit converges to 100% with linear complexity" Theorem 2 seems to cover this claim, so I am happy with this.

Claim 3: "the proposed method can accurately design realworld circuit modules comprising up to 5,000 logic gates". Here the writing is so poor that it is difficult for me to verify that this is true.

Table 1 is the key empirical artefact of the paper and needs much better explanation. Reviewers are NOT obliged to refer to Appendices, and a proper explanation is needed here. Even going to the Appendices I still struggle to understand your results.

How does this approach scale? I see you clear outputs that help me understand this crucial question.

Overall, the poor writing leaves me wondering what actually has been achieved. I don't see clear evidence to back up your claims.

**Questions:**

1. What is the reduction is state-space complexity of the approach?
2. Show formally of empirically how the approach scales.

---

> ### Author Response · Authors · 2025-11-20
>
> We sincerely thank the reviewer for the positive assessment of our experimental results in the summary and for the constructive feedback throughout the review. We are pleased that the empirical strengths of our work were recognized.
>
> Below, we provide a point-by-point response to the concerns raised. We are committed to addressing all issues thoroughly and welcome any further questions regarding our rebuttal.
>
> **Weakness 1:** The paper is very poorly written. The claims made are strong, but this reviewer has struggled to understand the technical basis for the claims, and to understand the empirical results.
>
> **Response 1:** Thanks for the advice to improve our paper. The key contribution of this paper is the Circuit HMM model structure, whose deterministic characteristic addresses the accuracy challenge in automated logic design. We propose that the key technical improvements lie in:  **(claim 1)** reducing complexity, **(claim 2)** ensuring high accuracy, and **(claim 3)** achieving better results than the state of the art on general datasets in the field. We much appreciate the accurate summary and will gladly modify the corresponding content in the paper.

---

> ### Author Response · Authors · 2025-11-20
>
> **Weakness 2: (about claim 1)** I don't understand the first main claim: "It formalizes the input-output relationships as a Markov Process, which reduces the complexity of the design space." Where is this claim (a) formally stated and (b) proven theoretically or empirically? Section 4.2 seems to be where this is done, but this is entirely inadequate. I am looking for direct, clear evidence that you reduce the complexity of the design space. By what factor is this reduction? Guaranteed reduction, or expected reduction? Your explanation of "complexity" is poor. I understand the complexity of combinational circuits; now the complexity of sequential circuits is different. Estimating the complexity of the design space for sequential circuits is a challenging problem because it involves both combinatorial and temporal dimensions. This article needs a clear exposition of this. Total Complexity ~ Combinational Topologies × State Machines, and if the number of distinct state machines with m states and i inputs is roughly: FSM Complexity ~ (2^m)^(2^i 2^m) This grows double-exponentially with the number of inputs and memory elements. How precisely does your approach reduce this complexity? What are the precise new bounds?
>
> **Response 2:** Thank you very much for your constructive suggestions. We are glad to clarify the concern about how our modeling reduces the complexity, and we will add this clarification to the revision. Given a sequential circuit design problem, let $DS_{temp}=DS(n,m,t)$ be the design space where the output depends on present input signals and the input sequence over past $t$ clock cycles (i.e., the temporal design space), and let $DS_{FSM}=DS(n,m,l)$ be the design space where the output depends only on present input and present state (i.e., the FSM design space), where $n$ is the bit width of primary inputs, $m$ is the bit width of primary outputs, and $l$ is the bit width of internal state registers. Then we have the Claim 1:
>
> > **Claim** (Design Space Complexity Reduction)
> > Given a sequential circuit design problem, to precisely model a sequential circuit's complete behavior, the traditional temporal modeling design space complexity is $DS_{temp}\\sim \\mathcal{O}(2^{m t\\cdot 2^{n t}})$, our FSM modeling design space complexity is $DS_{FSM}\\sim \\mathcal{O}(2^{(m+l)\\cdot 2^{n+l}})$, and for sequential circuits with at least one PI ($n \\geq 1$ and $l \\geq 1$), FSM modeling achieves a double-exponential reduction in design space complexity $\\sim \\mathcal{O}(2^{m\\cdot 2^{n\\cdot 2^{l}}})$ compared to temporal modeling.
> >
> > Note that the observation period must span at least $t \\geq 2^l$ clock cycles to precisely model a sequential circuit's complete behavior.
> >
> > **Proof**
> > For the complexity of the temporal design space, there are $2^{n t}$ possible input sequences over $t$ cycles, each requiring a mapping to one of $2^{m t}$ output sequences, yielding $2^{m t \\cdot 2^{n t}}$ distinct behaviors. For the complexity of the FSM design space, the transition and output functions must define behavior for all $2^{n+l}$ state-input combinations, with each combination mapping to one of $2^l$ next states and one of $2^m$ outputs, resulting in $2^{(m+l)\\cdot 2^{n+l}}$ possible implementations. Then we consider the ratio of design space complexities:
> >
> > $$\\frac{|DS_{temp}|}{|DS_{FSM}|} = \\frac{2^{m t \\cdot 2^{n t}}}{2^{(m+l) \\cdot 2^{n+l}}} = 2^{m t \\cdot 2^{n t} - (m+l) \\cdot 2^{n+l}}$$
> >
> > For $t \\geq 2^l$, we analyze the exponent $E$:
> >
> > $$E = m t \\cdot 2^{n t} - (m+l) \\cdot 2^{n+l}\\geq m\\cdot 2^{l+n\\cdot 2^l} - (m+l) \\cdot 2^{n+l}\\sim \\mathcal{O}(m\\cdot 2^{n\\cdot 2^{l}})$$
> >
> > Therefore, the exponent $E$ grows double-exponentially with respect to $l$, making the ratio $\\frac{|DS_{temp}|}{|DS_{FSM}|}$ triple-exponentially large ($\\sim \\mathcal{O}(2^{m\\cdot 2^{n\\cdot 2^{l}}})$). This demonstrates that FSM modeling achieves a triple-exponential reduction in design space complexity, as $|DS_{FSM}|$ is triple-exponentially smaller than $|DS_{temp}|$ for sequential circuits with $n \\geq 1$ and $l \\geq 1$.
>
> **Weakness 3: (about claim 2)**
> Claim 2: "we prove that the design accuracy of the sequential circuit converges to 100% with linear complexity" Theorem 2 seems to cover this claim, so I am happy with this.
>
> **Response 3:** We sincerely thank the reviewer for their positive assessment of Theorem 2 and their acknowledgment that it satisfactorily substantiates Claim 2 regarding the convergence to 100% accuracy with linear complexity.

---

> ### Author Response · Authors · 2025-11-20
>
> **Weakness 4: (about claim 3)** Claim 3: "the proposed method can accurately design real-world circuit modules comprising up to 5,000 logic gates". Here the writing is so poor that it is difficult for me to verify that this is true.
>
> Table 1 is the key empirical artefact of the paper and needs much better explanation. Reviewers are NOT obliged to refer to Appendices, and a proper explanation is needed here. Even going to the Appendices I still struggle to understand your results.
>
> How does this approach scale? I see you clear outputs that help me understand this crucial question.
>
> **Response 4:** Claim 3 is to demonstrate that the proposed method is scalable. Theoretically, as in Claim 1, the proposed method is linear-complex, so it is scalable (much more scalable than the state-of-the-art). Practically, we evaluate with experiments against the SOTA. The SOTA circuit design method (the deterministic method: BSD [1]) only achieves 60% accuracy on a circuit with 36 logic gates. Therefore, compared to SOTA, we have increased the scale by two orders of magnitude in accurately generating sequential circuits, pushing the scale limit of the automated logic design. In fact, circuits with fewer than 5000 logic gates are practical in the real world (a 32-bit full adder contains only approximately 160 logic gates, furthermore, our DW benchmark can cover most of the commonly used IPs), demonstrating that the proposed method is applicable.
>
> **Weakness 5:** Overall, the poor writing leaves me wondering what actually has been achieved. I don't see clear evidence to back up your claims.
>
> **Response 5:** We sincerely thank the reviewer for this overarching comment and for the detailed, constructive feedback provided throughout the review. In our responses above, we have provided a formal statement and proof of the complexity reduction of our modeling (Response 2) and given a detailed explanation of our empirical results and scalability argument (Response 4). We believe these detailed clarifications successfully resolve the core uncertainties raised. Based on this feedback, we are fully committed to incorporating all these explanations and improvements into a thoroughly revised manuscript. We are deeply grateful for the reviewer's insights and are readily available to address any further questions.
>
> **Question 1:** What is the reduction is state-space complexity of the approach?
>
> **Answer 1:** Our approach reduces the state-space complexity from the triple-exponential complexity of traditional temporal modeling $O(2^{m t \\cdot 2^{n t}})$ to the FSM modeling complexity $O(2^{(m+l) \\cdot 2^{n+l}})$, achieving a triple-exponential reduction as formally proven in **Response 2**.
>
> **Question 2:** Show formally of empirically how the approach scales.
>
> **Answer 2:** The approach scales linearly with the number of reachable states, as theoretically guaranteed by Theorem 2 **(Response 3)** and empirically demonstrated by our ability to accurately design sequential circuits with up to 5,000 logic gates **(Response 4)**. This represents a two-order-of-magnitude improvement over the state-of-the-art method BSD, which achieves only 60% accuracy on circuits with 36 logic gates, confirming both the theoretical linear complexity and practical scalability of our method.
>
> ## References
>
> [1] Cheng, S., Jin, P., Guo, Q., Du, Z., Zhang, R., Hu, X., Zhao, Y., Hao, Y., Guan, X., Han, H., et al. (2024). Automated CPU design by learning from input-output examples. In *Proceedings of the Thirty-Third International Joint Conference on Artificial Intelligence* (pp. 3843–3853).

---

### Official Review · Reviewer_NGYy · 2025-10-31

**Soundness:** 4
**Presentation:** 3
**Contribution:** 3
**Rating:** 6
**Confidence:** 2

**Summary:**

The authors propose a deterministic hidden Markov model for automatically generating a sequential circuit. Their method performs better than the deterministic method and the probabilistic hidden Markov method.

**Strengths:**

1. The performance of this work is much better than the baseline methods.
2. The modeling method of sequential behavior is insightful.
3. The complexity of this method is better than probabilistic methods.

**Weaknesses:**

1. It seems that only one level of latches will be considered in this method, which is not general enough for all types of sequential circuits.

**Questions:**

1. I am a little bit confused about the "deterministic HMM" that is claimed by the authors in the paper. In \textbf{State Mining} and \textbf{State Encoding}, "randomly generate a set of PIs" and "optimize the state encoding using a Simulated Annealing algorithm" could add randomness in the algorithm. Maybe the authors should explain the "deterministic" with more words.

---

> ### Author Response · Authors · 2025-11-20
>
> We sincerely thank the reviewer for the positive assessment of our work's performance and for the recognition of our method's insightful modeling approach for sequential behavior. We are particularly grateful for the acknowledgment that our method's performance is superior to baseline approaches and for noting its computational advantages.
>
> Below, we provide point-by-point responses to the specific concerns raised. We are readily available to address any further questions regarding our rebuttal.
>
> **Weakness 1:** It seems that only one level of latches will be considered in this method, which is not general enough for all types of sequential circuits.
>
> **Response 1:** It is actually a misunderstanding, because our method can handle multi-level circuits, not just one-level latch circuits. Although at each step of state mining we can only discover the state of the next-level latch, since this is an iterative process, we do implement the multi-level latch circuits. Of course, the more layers a latch has, the more difficult the design becomes.
>
> **Question 1:** I am a little bit confused about the "deterministic HMM" that is claimed by the authors in the paper. In **State Mining** and **State Encoding**, "randomly generate a set of PIs" and "optimize the state encoding using a Simulated Annealing algorithm" could add randomness in the algorithm. Maybe the authors should explain the "deterministic" with more words.
>
> **Answer 1:** We thank you for your kind advice and are glad to clarify. As stated in the paper (Line 137, **Definition 1**), the output of our method is a deterministic circuit rather than a probability distribution: that is, for any input sequence, its output is always consistent. On the contrary, for a probabilistic circuit, given the same input at different times, its output results exhibit randomness.
>
> > **Definition 1**(Automated Sequential Circuit Design)
> > There is a sequential logic oracle $\\phi : \\left\\{0, 1\\right\\}^n \\mapsto \\left\\{0, 1\\right\\}^m$, which can only be observed by the input-output examples. Given at most $N$ input-output probes from the oracle $\\left\\{\\left(\\mathbf{x}_1, \\phi(\\mathbf{x}_1)\\right), \\left(\\mathbf{x}_2, \\phi(\\mathbf{x}_2)\\right),\\right.$ $\\dots,$ $\\left.\\left(\\mathbf{x}_N, \\phi(\\mathbf{x}_N)\\right)\\right\\}$, where $\\mathbf{x}_i, \\phi(\\mathbf{x}_i)$ is the input/output sequence, generate a sequential circuit logic $\\psi$ to simulate $\\phi$,
> > such that $\\forall \\mathbf{x} \\in \\left\\{0, 1\\right\\}^n$,
> > $$P(\\phi (\\mathbf{x}) = \\psi (\\mathbf{x})) \\geq 1-\\epsilon ~(\\epsilon \\rightarrow 0),$$
> > where the sequential circuit $\\psi$ is a valid output design.
>
> The randomness in the method is used for efficiency, but does not affect the output quality. There are multiple deterministic sequential circuits that meet the given requirement, and our algorithm can output any such circuit. Therefore, we relaxed the constraint that the algorithm must output the same circuit every time, as long as it comes from this set. Therefore, heuristic methods and random sampling can be used to reduce complexity further. Moreover, if the user needs the design process to be deterministic, we only need to set a fixed random seed for reproducibility.

---

### Official Review · Reviewer_uoii · 2025-10-31

**Soundness:** 2
**Presentation:** 2
**Contribution:** 2
**Rating:** 4
**Confidence:** 4

**Summary:**

This paper is an extension of Binary Speculation Diagram Learner (Cheng, 2024a), from combinatorial to sequential circuit design. Given a black-box sequential circuit (oracle), the proposed algorithm aims to implement it. The algorithm firstly uses breadth-first search to find all the internal states of the sequential circuit, which iteratively finds new reachable states from current known states (by randomly generated PIs). Then all the $N$ states are encoded as $⌈log_2 N⌉$ bits. In this way the problem reduces to $N$ combinatorial circuit design problems which can be solved by (Cheng, 2024a). The state encoding can be optimized by a simulated annealing algorithm which minimizes the BSD node count. Experimental result shows that it can successfully generate sequential circuits up to thousands of states and gates.

**Strengths:**

- This paper focuses on generating sequential circuits, which is more challenging and less explored in literatures than combinatorial ones
- The empirical experimental result is good
- Many details in the appendix

**Weaknesses:**

- The idea of regarding sequential logic circuits as Markov processes is not very new (if it is not a common sense). Sequential logic circuits can be modelled as finite state machines, so Figure 1 (c) is actually more aligned with my knowledge on sequential circuits, and the perspective shown in Figure 1 (b) may not be so common.
- The problem definition (section 2.1) confused me. While it is directly referenced from (Cheng, 2024a), I feel that the actual problem in this paper is different. The input is not a fixed dataset of $N$ input-output pairs, but rather a black-box sequential circuit, which can be interacted by any inputs during the algorithmic process. I think only in such a way can the random generation of PIs (line 252) make sense.
- The impact of the paper is not clearly presented. Given that the target sequential circuit does exist and can be interacted, the paper is more likely a "reverse engineering" of an existing sequential circuit design, rather than learning a design from scratch based on examples of input-output pairs. It is not clear to me how the proposed approach can be integrated into existing EDA tools (line 1014).
- The comparison with baselines methods (Transformer, LSTM, BSD) is not very fair. The proposed approach can interact with the oracle with arbitrary inputs (online), while the other approaches learn on a fixed dataset (offline). I think the main contribution of this paper is the state mining algorithm, so a stronger baseline would be to replace the proposed state mining algorithm with other approaches (including those mentioned in appendix A.7) and compare the final accuracy and circuit size.

**Questions:**

- I wonder whether a sequential logic circuit can be defined as a function $\phi:\\{0,1\\}^n\rightarrow\\{0,1\\}^m$ (line 140). Due to the existence of states, the output of a circuit can be different even if the same inputs are fed into the circuit.

---

> ### Author Response · Authors · 2025-11-20
>
> We sincerely thank the reviewer for the thoughtful review and positive comments regarding our strong experimental results and the comprehensive details provided in the appendix. We are pleased that you found the empirical evaluation to be solid.
>
> In this rebuttal, we provide a point-by-point response to the concerns raised. We are happy to address any further questions the reviewer may have.
>
> **Weakness 1:**  The idea of regarding sequential logic circuits as Markov processes is not very new (if it is not a common sense). Sequential logic circuits can be modeled as finite state machines, so Figure 1 (c) is actually more aligned with my knowledge on sequential circuits, and the perspective shown in Figure 1 (b) may not be so common.
>
> **Response 1:**  We agree that HMMs are commonly used to describe sequential logic circuits, however, they cannot yet be used to design them. The key insight of our work is not only in modeling circuits as Markov processes, but also in formulating a deterministic HMM that can automate the design of these circuits from I/O examples. The probabilistic nature of existing HMMs makes them fundamentally incompatible with circuit synthesis, as their output distributions cannot be mapped to the precise Boolean functions required for a deterministic netlist. Consequently, state-of-the-art automated circuit design methods have exclusively relied on end-to-end methods such as neural networks (e.g., LLMs, Transformers) [1,2,3], without HMM models. Therefore, our primary academic contribution is the Circuit HMM model structure, which is distinguished by its deterministic nature and can be used to design sequential logic circuits with any small error bound $\\epsilon$, as proven in **Theorem 2**.
>
> **Weakness 2:** The problem definition (section 2.1) confused me. While it is directly referenced from (Cheng, 2024a), I feel that the actual problem in this paper is different. The input is not a fixed dataset of $N$ input-output pairs, but rather a black-box sequential circuit, which can be interacted by any inputs during the algorithmic process. I think only in such a way can the random generation of PIs (line 252) make sense.
>
> **Response 2:** It is a misunderstanding here. In the problem definition (both in **Section 2.1** and Cheng2024a [3]), the input is not a fixed set of IOs but the complete set of IOs. Under this problem description, designers can use all the IOs for automatic design. It is similar to the human design process, where engineers require the entire functional specification of the sequential circuit (rather than just a fixed part). In practice, when designing at a large scale (even with only 40 input bits), using all IOs for training is not realistic, as it requires TBs of training data. In this way, both our proposed method and the existing methods can only use part of the complete set, but it is not a fixed set given in advance.
>
> [1] Fu, Y., Zhang, Y., Yu, Z., Li, S., Ye, Z., Li, C., Wan, C., & Lin, Y. C. (2023). Gpt4aigchip: Towards next-generation ai accelerator design automation via large language models. In *2023 IEEE/ACM International Conference on Computer Aided Design (ICCAD)* (pp. 1–9). IEEE.
>
> [2] Li, X., Li, X., Chen, L., Zhang, X., Yuan, M., & Wang, J. (2024). Circuit Transformer: End-to-end Circuit Design by Predicting the Next Gate. *CoRR*.
>
> [3] Cheng, S., Jin, P., Guo, Q., Du, Z., Zhang, R., Hu, X., Zhao, Y., Hao, Y., Guan, X., Han, H., et al. (2024). Automated CPU design by learning from input-output examples. In *Proceedings of the Thirty-Third International Joint Conference on Artificial Intelligence* (pp. 3843–3853).

---

> ### Author Response · Authors · 2025-11-20
>
> **Weakness 3:** The impact of the paper is not clearly presented. Given that the target sequential circuit does exist and can be interacted, the paper is more likely a "reverse engineering" of an existing sequential circuit design, rather than learning a design from scratch based on examples of input-output pairs. It is not clear to me how the proposed approach can be integrated into existing EDA tools (line 1014).
>
> **Response 3:** We agree that the method can be applied to reverse engineering, but we emphasize that its scope and impact are significantly broader. The capability to automatically synthesize a circuit from a functional oracle is foundational. This oracle can indeed be an existing circuit for reverse engineering, but more importantly, it can be a high-level model, a software simulator, or any executable specification for forward design automation. This allows our method to automatically generate RTL from a specification, a task traditionally performed by human designers.
>
> Regarding integration into the EDA flow, our work establishes automatic logic design as a new subflow. This subflow expands the starting point of the contemporary EDA process in the AI era. It automatically generates RTL code for the circuit design from the functional specification (e.g., a truth table), which is conventionally considered the domain of human designers. The current mainstream EDA flow starts with logic synthesis/optimization, which is now the second stage: the output Verilog code from automated logic design (such as from the Circuit HMM) serves as the input to logic synthesis/optimization, replacing the human-designed RTL code. The current EDA flow can further optimize automated logic design in an orthogonal manner.
>
> **Weakness 4:** The comparison with baselines methods (Transformer, LSTM, BSD) is not very fair. The proposed approach can interact with the oracle with arbitrary inputs (online), while the other approaches learn on a fixed dataset (offline).
>
> **Response 4:** We acknowledge the difference in learning paradigm (online vs. offline). However, our comparison is fair in terms of the fundamental resource for learning: the input-output (IO) data from the oracle.
> The misunderstanding likely stems from the problem definition clarified in **Weakness 2 & Response 2**. Crucially, both our method and all baseline methods utilize the same underlying source of information: the complete set of IO pairs from the black-box oracle. To ensure a fair and quantitative comparison, the training set for the end-to-end NNs (Transformer, LSTM) consists of a large, fixed dataset of 100,000 IO examples, each a 100-length sequence. And our proposed method, in its state mining phase, uses an average of 500 IO examples per discovered state. Given the average number of states in our benchmarks is 130.05, the total IO examples used by our method is less than the 100,000 provided to the baselines. Moreover, the baseline methods are much slower than the proposed method and thus are not suitable for more IO examples.

---

> ### Author Response · Authors · 2025-11-20
>
> **Weakness 5:** I think the main contribution of this paper is the state mining algorithm, so a stronger baseline would be to replace the proposed state mining algorithm with other approaches (including those mentioned in appendix A.7) and compare the final accuracy and circuit size.
>
> **Response 5:** We disagree that our contribution is the state mining algorithm itself. The key contribution of this paper is the Circuit HMM model structure, whose deterministic characteristic addresses the accuracy challenge in automated logic design. Therefore, we compared this task with SOTA, and believe that the comparison with other state mining methods is an ablation study for effectiveness, rather than a complete comparison (see Appendix 7).
>
> We thank you for your kind advice and have made further comparison in the table below. In our design process, the accuracy of not using state mining is 60.03% (Section 5.1), the average accuracy of using the SOTA method ( L*& L#) for state mining is 34.89% & 34.37% (we only show 10/42 circuits where SOTA methods fail to find the correct states), and the average accuracy of using our method for state mining is 100.00%. This not only demonstrates the effectiveness of our algorithm (as detailed in Appendix A.7 for data comparison), but also shows that our design flow is necessary and effective for addressing the critical issue of insufficient accuracy in the automatic design of sequential circuits.
>
> **Table: Accuracy Comparison of State Mining with SOTA**
>
> | Circuit Name    | L*      | L#      | Ours     |
> |-----------------|---------|---------|----------|
> | Prob121         | 80.00%  | 80.00%  | 100.00%  |
> | Prob146         | 33.39%  | 33.39%  | 100.00%  |
> | Prob155         | 25.00%  | 20.45%  | 100.00%  |
> | bictr_scnto     | T/o     | T/o     | 100.00%  |
> | lfsr_updn       | 96.47%  | 95.29%  | 100.00%  |
> | updn_ctr        | T/o     | T/o     | 100.00%  |
> | arb_rr          | 25.00%  | 25.00%  | 100.00%  |
> | cntr_gray       | T/o     | T/o     | 100.00%  |
> | pulse_sync      | 20.75%  | 22.64%  | 99.99%   |
> | pulseack_sync   | 68.24%  | 66.89%  | 100.00%  |
> | **Average**     | 34.89%  | 34.37%  | 100.00%  |
>
> *Note: T/o indicates timeout (exceeded 6-hour time limit).*
>
> **Question 1:** I wonder whether a sequential logic circuit can be defined as a function $\\phi : \\left\\{0, 1\\right\\}^n \\mapsto \\left\\{0, 1\\right\\}^m$ (line 140). Due to the existence of states, the output of a circuit can be different even if the same inputs are fed into the circuit.
>
> **Answer 1:** We sincerely thank the reviewer for their careful reading and for identifying this important conceptual error. We agree that a sequential logic circuit cannot be described as a function due to its state-dependent behavior. We acknowledge this mistake and have made the following corrections throughout the manuscript:
>
> > **Definition 1**(Automated Sequential Circuit Design)
> > There is a sequential logic oracle $\\phi : \\left\\{0, 1\\right\\}^n \\mapsto \\left\\{0, 1\\right\\}^m$, which can only be observed by the input-output examples. Given at most $N$ input-output probes from the oracle $\\left\\{\\left(\\mathbf{x}_1, \\phi(\\mathbf{x}_1)\\right), \\left(\\mathbf{x}_2, \\phi(\\mathbf{x}_2)\\right),\\right.$ $\\dots,$ $\\left.\\left(\\mathbf{x}_N, \\phi(\\mathbf{x}_N)\\right)\\right\\}$, where $\\mathbf{x}_i, \\phi(\\mathbf{x}_i)$ is the input/output sequence, generate a sequential circuit logic $\\psi$ to simulate $\\phi$,
> > such that $\\forall \\mathbf{x} \\in \\left\\{0, 1\\right\\}^n$,
> > $$P(\\phi (\\mathbf{x}) = \\psi (\\mathbf{x})) \\geq 1-\\epsilon ~(\\epsilon \\rightarrow 0),$$
> > where the sequential circuit $\\psi$ is a valid output design.

---

> ### Comment · Reviewer_uoii · 2025-11-23
>
> Thanks for the clarification and further information.
>
> For response 1 & 3, you may include the clarification in the paper (main text or appendix), which helps readers understand the motivation.
>
> For response 2 & 4, I am still a bit confused. You say that the input is the complete set of IOs and designers can use all the IOs for automatic design. However, in line 141 of the paper, it says
> > Given at most $N$ input-output probes from the oracle ${(x_1, φ(x_1)) , (x_2, φ(x_2)) , . . . , (x_N , φ(x_N ))}$, ...
>
> The way you are writing now gives the impression that a fixed dataset of $N$ input-output pairs ${(x_1, φ(x_1)) , (x_2, φ(x_2)) , . . . , (x_N , φ(x_N ))}$ is already there before the algorithmic process. I guess what you may actually want to express is that
>
> > Given at most $N$ **chances** of input-output probes ${(x_1, φ(x_1)) , (x_2, φ(x_2)) , . . . , (x_N , φ(x_N ))}$ from the oracle, ...
>
> In such a way, the readers will know that the data is not present before the algorithmic process.
>
> I also wonder what you mean by "the complete set of IOs". If you mean all possible input-output pairs, its number would be $2^N \times |S|$ in which $S$ is the set of all the states. This is typically a huge number which is much larger than 100,000.
>
> For response 5, maybe I should rephrase my review as "I think the state mining algorithm is the most technically challenging part of the paper" (sec 4.1 compared with 4.2 and 4.3), sorry for any misunderstanding. Can you provide more details about how you use AALpy mentioned in line 968 for state mining? (You may simply attach a code snippet in the supplementary material)

---

> ### Author Response · Authors · 2025-12-03
> **Further Reply**
>
> We are glad our previous responses were helpful. Here we concisely address the two remaining points regarding the “complete set of IOs” and the use of AALpy.
>
> **On the term “complete set of IOs”:**
> We clarify that this phrase simply means all compared methods operate under the same online-query setting: each can request the output for any chosen input sequence from the oracle, subject only to a total query budget $N$. This contrasts with learning from a fixed offline dataset, and the resource constraint is the number of queries, not accessibility of the I/O space.
>
> **On the use of AALpy for baseline comparison:**
> AALpy requires a System Under Learning (SUL), which is a black-box interface with reset and step functions, to learn a Deterministic Finite Automaton (DFA), i.e., a state-transition table. We wrapped our circuit oracle into such a SUL and for evaluation, we generated 10,000 random input sequences of length 100 and executed them in parallel on the original oracle and the learned state machine. The following code snippet illustrates the integration (which we will add to the supplement):
>
> ```python
>   sul_lstar = ExternalMealySUL(lib, PI_WIDTH, LATCH_WIDTH, PO_WIDTH)
>   eq_oracle_lstar = RandomWalkEqOracle(input_alphabet, sul_lstar, num_steps=1000)
>
>   learned_lstar = run_Lstar(
>       alphabet=input_alphabet,
>       sul=sul_lstar,
>       eq_oracle=eq_oracle_lstar,
>       automaton_type='mealy',
>       cex_processing='rs',
>       print_level=1
>    )
>   sul_lsharp = ExternalMealySUL(lib, PI_WIDTH, LATCH_WIDTH, PO_WIDTH)
>   eq_oracle_lsharp = RandomWalkEqOracle(input_alphabet, sul_lsharp, num_steps=1000)
>
>   learned_lsharp = run_Lsharp(
>       alphabet=input_alphabet,
>       sul=sul_lsharp,
>       eq_oracle=eq_oracle_lsharp,
>       automaton_type='mealy',
>       separation_rule="SepSeq",
>       max_learning_rounds=2000,
>       print_level=1
>    )
> ```
>
> The critical distinction lies in our state mining strategy and validation mechanism, which are specifically designed for circuit design. AALpy's L*/L# algorithms rely on a minimal adequate teacher model, requiring exact counterexamples—a process that is computationally expensive and sensitive to oracle behavior. In contrast, our method employs a BFS-based exploration guided by efficient, output-driven state validation. We use a probabilistic equivalence check based on long, random input sequences instead of seeking perfect counterexamples. This approach is more robust and scalable for circuit oracles, trading the cost of guaranteed minimality for a massive reduction in query complexity. Consequently, our state mining not only discovers states more effectively but also produces a state set directly suitable for the subsequent stages of state encoding and circuit generation, which is the core of our contribution.
>
> We will incorporate these clarifications into the final manuscript.

---

### Author Response · Authors · 2025-12-03
**A Concise Summary of Rebuttal**

**Dear Area Chair,**

Thank you for efficiently handing our submission and leading the rebuttal discussion. To assist your final assessment, we provide a concise summary of our rebuttal and the additional experiments conducted to address all concerns.

**1. Summary of Strengths**

Reviewers highlighted several core strengths of our work:
* **Strong Experimental Performance & Scalability:** The empirical experimental result is strong (uoii, y3kG, NGYy), and demonstrating clear advantages in terms of accuracy, scale, and efficiency (HFWt).
* **Grounded Theorem & Insightful Method:** The modeling method of sequential behavior is insightful, resulting a better complexity than probabilistic methods (NGYy), and the theoretical derivations are logically rigorous (HFWt).
* **Well-Structured Presentation & Practical Relevance:**
The presentation has many details in the appendix (uoii) and addresses the important problem of automated sequential circuit design, offering well-defined practical value (HFWt).

**2. Summary of Rebuttal & Clarifications**

The reviewers' primary concerns involved novelty, Soundness, Scope, ablation and clarity. We addressed each via the following revisions and experiments:
* **Novelty (uoii, HFWt):** We clarified that the novelty of our work lies not in inventing the Markov model itself, but in formulating a deterministic, design-oriented HMM framework that bridges the gap between functional specification (I/O behavior) and correct circuit implementation, a step that prior probabilistic HMMs could not support. We have refined the description of our framework in the paper to better distinguish it from pure modeling techniques and to highlight its role in enabling automated sequential circuit design.
* **Definition Clarity(uoii):** We clarified that the input is access to the complete I/O behavior of the oracle, not a fixed dataset, aligning with real-world design scenarios where functional specifications are fully observable.
* **Comparison Fairness (uoii, HFWt):** We demonstrated that all methods use the same underlying I/O data source, and our method often uses fewer examples while achieving superior accuracy and speed.
* **Technical Scope & Soundness (NGYy, y3kG):** We (a) proved a triple-exponential reduction in design space complexity, (b) clarified support for multi-level circuits, and (c) explained that “deterministic” refers to circuit output, not the search algorithm.
* **Ablation Studies (HFWt, uoii):** We showed our state encoding optimization reduces circuit size by ~85.5% on average, and our state mining achieves 100% accuracy vs. ~35% for SOTA alternatives.
* **Writing Clarity (y3kG):** we have updated the explanation of Table 1 to make the empirical results more accessible. Crucially, we have also added a formal proof (Claim 1) detailing how our FSM modeling reduces design space complexity compared to traditional temporal modeling, directly addressing the request for theoretical evidence.

**3. Summary of Committed Revisions**

We updated the manuscript to incorporate these results and analyses:
- **Expanded Experiments:** Added comprehensive ablation studies on state encoding and state mining.
- **New Appendices:**
  - Formally stated Claim 1 and provided its complete proof in the Appendix A.3.1.
  - Detailed baseline parameters and training setups in the Appendix A.5.3.
  - Updated the explanation of Table 1 to make the empirical results more accessible in Appendix A.6.
  - Comparative analysis with other state mining methods (e.g., L* & L#)in the Appendix A.7.
  - Results for the state encoding ablation study A.8.

* **Minor Corrections:** Fixed Theorem 1.

We believe our detailed rebuttal has fully resolved the reviewers' concerns. We are committed to delivering a revised manuscript that reflects these improvements and underscores the significant advance of our work. Thank you for your time and consideration.

**Sincerely,
The Authors of Submission 11493**

---

### Meta-Review · Area_Chair_ycyA · 2026-01-06

**Summary:**

This paper proposes "Circuit HMM," a deterministic Hidden Markov Model framework for automating the design of sequential logic circuits. The method treats input-output relationships as a Markov Process to reduce design space complexity. It utilizes a three-stage pipeline: heuristic state mining via BFS, state encoding optimization via simulated annealing, and combinational logic synthesis using a Binary Speculation Diagram (BSD) learner.  A primary reason for rejection is the poor quality of writing and presentation, which significantly hinders technical verification. Reviewer y3kG noted that "the writing is so poor that it is difficult... to verify that this is true" and found the initial explanations of core claims (e.g., complexity reduction) "entirely inadequate". Table 1, a key artifact, was criticized for lacking sufficient explanation. Another reason is the limited novelty: Reviewers HFWt and uoii pointed out a lack of fundamental algorithmic innovation. The "Circuit HMM" is essentially a deterministic adaptation of classical HMM frameworks, while the core components—state mining (BFS/Monte Carlo) and state encoding (Simulated Annealing)—are applications of well-established algorithms rather than novel contributions. In addition, Reviewer uoii raised valid concerns regarding the fairness of comparing the proposed online method (which interacts arbitrarily with an oracle) against offline baselines trained on fixed datasets. Furthermore, the method is validated solely on synchronous circuits, failing to address asynchronous designs or those with complex feedback loops, limiting the generalizability of the findings.

**Reviewer Concerns:**

1. Theoretical Basis for Complexity Reduction (Addressed): Reviewer y3kG challenged the authors' core claim that formalizing the problem as a Markov Process reduces design space complexity, initially finding the explanation "entirely inadequate." In the rebuttal, the authors successfully addressed this by providing a formal proof ("Claim 1") demonstrating that their FSM modeling achieves a triple-exponential reduction in design space

2. Manuscript Readability and Clarity (Outstanding): Reviewer y3kG rated the paper as a "reject" (score of 2) largely because "the writing is so poor that it is difficult... to verify that this is true," making it hard to understand the technical basis and empirical results. While the authors acknowledged this and committed to revising the text to incorporate their clearer rebuttal explanations, the readability of the final manuscript remains a critical outstanding hurdle that dictates whether the technical contributions can be properly assessed and appreciated.

**Reviewer Scores:**

The first and third reviewers might increase their scores by 2. The first reviewer explicitly acknowledged the rebuttal was helpful, despite also raising some other concerns. The third reviewer asked for a theoretical justification which was provided by the authors.

---

### Decision · Program_Chairs · 2026-01-26

Reject